# A population-level invasion by transposable elements triggers genome expansion in a fungal pathogen

Ursula Oggenfuss[1], Thomas Badet[1], Thomas Wicker[2], Fanny E Hartmann[3,4], Nikhil Kumar Singh[1], Leen Abraham[1], Petteri Karisto[4†], Tiziana Vonlanthen[4], Christopher Mundt[5], Bruce A McDonald[4], Daniel Croll[1]*

[1]Laboratory of Evolutionary Genetics, Institute of Biology, University of Neuchâtel, Neuchatel, Switzerland; [2]Institute for Plant and Microbial Biology, University of Zurich, Zurich, Switzerland; [3]Ecologie Systématique Evolution, Bâtiment 360, Univ. Paris-Sud, AgroParisTech, CNRS, Université Paris-Saclay, Orsay, France; [4]Plant Pathology, Institute of Integrative Biology, ETH Zurich, Zurich, Switzerland; [5]Department of Botany and Plant Pathology, Oregon State University, Corvallis, United States

*For correspondence:
daniel.croll@unine.ch

Present address: †Plant Health, Natural Resources Institute Finland (Luke), Jokioinen, Finland

Competing interest: The authors declare that no competing interests exist.

**Abstract** Genome evolution is driven by the activity of transposable elements (TEs). The spread of TEs can have deleterious effects including the destabilization of genome integrity and expansions. However, the precise triggers of genome expansions remain poorly understood because genome size evolution is typically investigated only among deeply divergent lineages. Here, we use a large population genomics dataset of 284 individuals from populations across the globe of *Zymoseptoria tritici*, a major fungal wheat pathogen. We built a robust map of genome-wide TE insertions and deletions to track a total of 2456 polymorphic loci within the species. We show that purifying selection substantially depressed TE frequencies in most populations, but some rare TEs have recently risen in frequency and likely confer benefits. We found that specific TE families have undergone a substantial genome-wide expansion from the pathogen's center of origin to more recently founded populations. The most dramatic increase in TE insertions occurred between a pair of North American populations collected in the same field at an interval of 25 years. We find that both genome-wide counts of TE insertions and genome size have increased with colonization bottlenecks. Hence, the demographic history likely played a major role in shaping genome evolution within the species. We show that both the activation of specific TEs and relaxed purifying selection underpin this incipient expansion of the genome. Our study establishes a model to recapitulate TE-driven genome evolution over deeper evolutionary timescales.

## Introduction

Transposable elements (TEs) are mobile repetitive DNA sequences with the ability to independently insert into new regions of the genome. TEs are major drivers of genome instability and epigenetic change (*Eichler and Sankoff, 2003*). Insertion of TEs can disrupt coding sequences, trigger chromosomal rearrangements, or alter expression profiles of adjacent genes (*Lim, 1988*; *Petrov et al., 2003*; *Slotkin and Martienssen, 2007*; *Hollister and Gaut, 2009*; *Oliver et al., 2013*). Hence, TE activity can have phenotypic consequences and impact host fitness. While TE insertion dynamics are driven by the selfish interest for proliferation, the impact on the host can range from beneficial to highly deleterious. The most dramatic examples of TE insertions underpinned rapid adaptation of populations or species (*Feschotte, 2008*; *Chuong et al., 2017*), particularly

following environmental change or colonization events. Beneficial TE insertions are expected to experience strong positive selection and rapid fixation in populations. However, most TE insertions have neutral or deleterious effects upon insertions. Purifying selection is expected to rapidly eliminate deleterious insertions from populations unless constrained by genetic drift (*Walser et al., 2006*; *Baucom et al., 2009*; *Cridland et al., 2013*; *Stuart et al., 2016*; *Lai et al., 2017*; *Stritt et al., 2018*). Additionally, genomic defense mechanisms can disable transposition activity. Across eukaryotes, epigenetic silencing is a shared defense mechanism against TEs (*Slotkin and Martienssen, 2007*). Fungi evolved an additional and highly specific defense system introducing repeat-induced point (RIP) mutations into any nearly identical set of sequences. The relative importance of demography, selection, and genomic defenses determining the fate of TEs in populations remain poorly understood.

A crucial property predicting the invasion success of TEs in a genome is the transposition rate. TEs tend to expand through family-specific bursts of transposition followed by prolonged phases of transposition inactivity. Bursts of insertions of different retrotransposon families were observed across eukaryotic lineages including *Homo sapiens*, *Zea mays*, *Oryza sativa*, and *Blumeria graminis* (*Shen et al., 1991*; *SanMiguel et al., 1998*; *Eichler and Sankoff, 2003*; *Piegu et al., 2006*; *Lu et al., 2017*; *Frantzeskakis et al., 2018*). Prolonged bursts without effective counterselection are thought to underpin genome expansions. In the symbiotic fungus *Cenococcum geophilum*, the burst of TEs resulted in a dramatically expanded genome compared to closely related species (*Peter et al., 2016*). Similarly, a burst of a TE family in brown hydras led to an approximately threefold increase of the genome size compared to related hydras (*Wong et al., 2019*). Across the tree of life, genome sizes vary by orders of magnitude and enlarged genomes invariably show hallmarks of historic TE invasions (*Kidwell, 2002*). Population size variation is among the few correlates of genome size across major groups, suggesting that the efficacy of selection plays an important role in controlling TE activity (*Lynch, 2007*). Reduced selection efficacy against deleterious TE insertions is expected to lead to a ratchet-like increase in genome size. In fungi, TE-rich genomes often show an isochore structure alternating gene-rich and TE-rich compartments (*Rouxel et al., 2011*). TE-rich compartments often harbor rapidly evolving genes such as effector genes in pathogens or resistance genes in plants (*Raffaele and Kamoun, 2012*; *Jiao and Schneeberger, 2019*). Taken together, incipient genome expansions are likely driven by population-level TE insertion dynamics.

The fungal wheat pathogen, *Zymoseptoria tritici,* is one of the most important pathogens on crops, causing high yield losses in many years (*Torriani et al., 2015*). *Z. tritici* emerged during the domestication of wheat in the Fertile Crescent where the species retained high levels of genetic variation (*Zhan et al., 2005*; *Stukenbrock et al., 2011*). The pathogen migrated to all temperate zones where wheat is currently grown and underwent multiple migration bottlenecks, in particular when colonizing Oceania and North America (*Zhan et al., 2005*; *Estep et al., 2015*). The genome is completely assembled and shows size variation between individuals sampled across the global distribution range (*Feurtey et al., 2020*; *Badet et al., 2020*; *Goodwin et al., 2011*). The TE content of the genome shows a striking variation of 17–24% variation among individuals (*Badet et al., 2020*). *Z. tritici* recently gained major TE-mediated adaptations to colonize host plants and tolerate environmental stress (*Omrane et al., 2015*; *Omrane et al., 2017*; *Krishnan et al., 2018*; *Meile et al., 2018*). Clusters of TEs are often associated with genes encoding important pathogenicity functions (i.e. effectors), recent gene gains or losses (*Hartmann and Croll, 2017*), and major chromosomal rearrangements (*Croll et al., 2013*; *Plissonneau et al., 2016*). Transposition activity of TEs also had a genome-wide impact on gene expression profiles during infection (*Fouché et al., 2019*). The well-characterized demographic history of the pathogen and evidence for recent TE-mediated adaptations make *Z. tritici* an ideal model to recapitulate the process of TE insertion dynamics, adaptive evolution, and changes in genome size at the population level.

Here, we retrace the population-level context of TE insertion dynamics and genome size changes across the species range by analyzing populations sampled on four continents for a total of 284 genomes. We developed a robust pipeline to detect newly inserted TEs using short read sequencing datasets. Combining analyses of selection and knowledge of the colonization history of the pathogen, we tested whether population bottlenecks were associated with substantial changes in the TE content and the size of genomes.

## Results

### A dynamic TE landscape shaped by strong purifying selection

We detected 4753 TE copies, grouped into 30 families with highly variable copy numbers in the reference genome IPO323 (*Figure 2—source data 1* and *Figure 2—figure supplement 1A*). To establish a comprehensive picture of within-species TE dynamics, we analyzed 295 genomes from a worldwide set of six populations spanning the distribution range of the wheat pathogen *Z. tritici*. To ascertain the presence or absence of TEs across the genome, we developed a robust pipeline (*Figure 1A*). In summary, we called TE insertions by identifying reads mapping to both a TE sequence and a specific location in the reference genome. Then, we assessed the minimum sequencing coverage to reliably recover TE insertions and removed 11 genomes with an average read depth below 15× (*Figure 1B*). We tested for evidence of TEs using read depth at target site duplications (*Figure 1C*) and scanned the genome for mapped reads indicating gaps at TE loci (*Figure 1D*). We found robust evidence for a total of 18,864 TE insertions grouping into 2465 individual loci. Of these loci, 35.5 % (n = 876) have singleton TEs (i.e., this locus is only present in one isolate: *Figure 2A*, *Figure 2—source data 3*). An overwhelming proportion of loci (2345 loci or 95.1%) have a TE frequency below 1 %. Singleton TE insertions in particular can be the product of spurious Illumina read mapping errors (*Nakamura et al., 2011*). To assess the reliability of the detected singletons, we focused on seven isolates for which PacBio long-read data was available (*Badet et al., 2020*). Aligned PacBio reads confirmed the exact location of 71 % (22 of 31 singleton insertions among seven isolates; see Materials and methods for further details). We found no significant difference in read coverage between confirmed and unconfirmed singleton insertions (*Figure 2—figure supplement 1B,C* and *Figure 2—source data 2*).

The abundance of singleton TE insertions strongly supports the idea that TEs actively copy into new locations but also indicates that strong purifying selection maintains nearly all TEs at low frequency (*Figure 2A*). The density of TE loci on accessory chromosomes, which are not shared among all isolates of the species, is almost twice the density found on core chromosomes (102 vs 58 TEs per Mb; *Figure 2B*, *Figure 2—figure supplement 2A*). This suggests relaxed selection against TE insertion on the functionally dispensable and gene-poor accessory chromosomes. We found no difference in TE allele frequency distribution between recombination hotspots and the rest of the genome (*Figure 2—figure supplement 2B*). Similarly, the TE density and the number of insertions did not vary between recombination hotspots and the genomic background (*Figure 2—figure supplement 2C*).

TEs grouped into 23 families and 11 superfamilies, with 88.2 % of all copies belonging to class I/retrotransposons (n = 2175; *Figure 2C*, *Figure 2—figure supplement 3A,B*). RLG/*Gypsy* (n = 1483) and RLC/*Copia* (n = 623) elements constitute the largest long terminal repeats (LTR) superfamilies. Class II/DNA transposons are dominated by DHH/*Helitron* (n = 249). As expected, TE families shared among fewer isolates tend to show also lower global copy numbers (i.e., all isolates combined), while TE families that are present in all isolates generally have high global copy numbers (*Figure 2D*).

We detected 153 loci with TEs inserted into genes with most of the insertions being singletons (44.7 %; n = 68) or of very low frequency (*Figure 2E*). Overall, TE insertions into exonic sequences were less frequent than expected compared to insertions into up- and downstream regions, which is consistent with effective purifying selection (*Figure 2F*). Insertions into introns were also strongly under-represented, likely due to the small size of most fungal introns (~50–100 bp) and the high probability of disrupting splicing or adjacent coding sequences. We also found that insertions 800–1000 bp away from coding sequences of a focal gene were under-represented. Given the high gene density, with an average spacing between genes of 1.744 kb, TE insertions within 800–1000 bp of a coding gene tend to be near adjacent genes already. Taken together, TEs in the species show a high degree of transposition activity and are subject to strong purifying selection.

### Detection of candidate TE loci underlying recent adaptation

The TE transposition activity can generate adaptive genetic variation. To identify the most likely candidate loci, we analyzed insertion frequency variation among populations as an indicator for recent selection. Across all populations, the insertion frequencies differed only weakly with a strong skew toward extremely low $F_{ST}$ values (mean = 0.0163; *Figure 3A,B*, *Figure 3—figure supplement 1*). To further analyze evidence for TE-mediated adaptive evolution, we screened a genome-wide SNP dataset for evidence of selective sweeps using selection scans. We found 16.5 % of all TE loci located in regions of selective sweep. Given our population sampling of two population pairs, we tested for

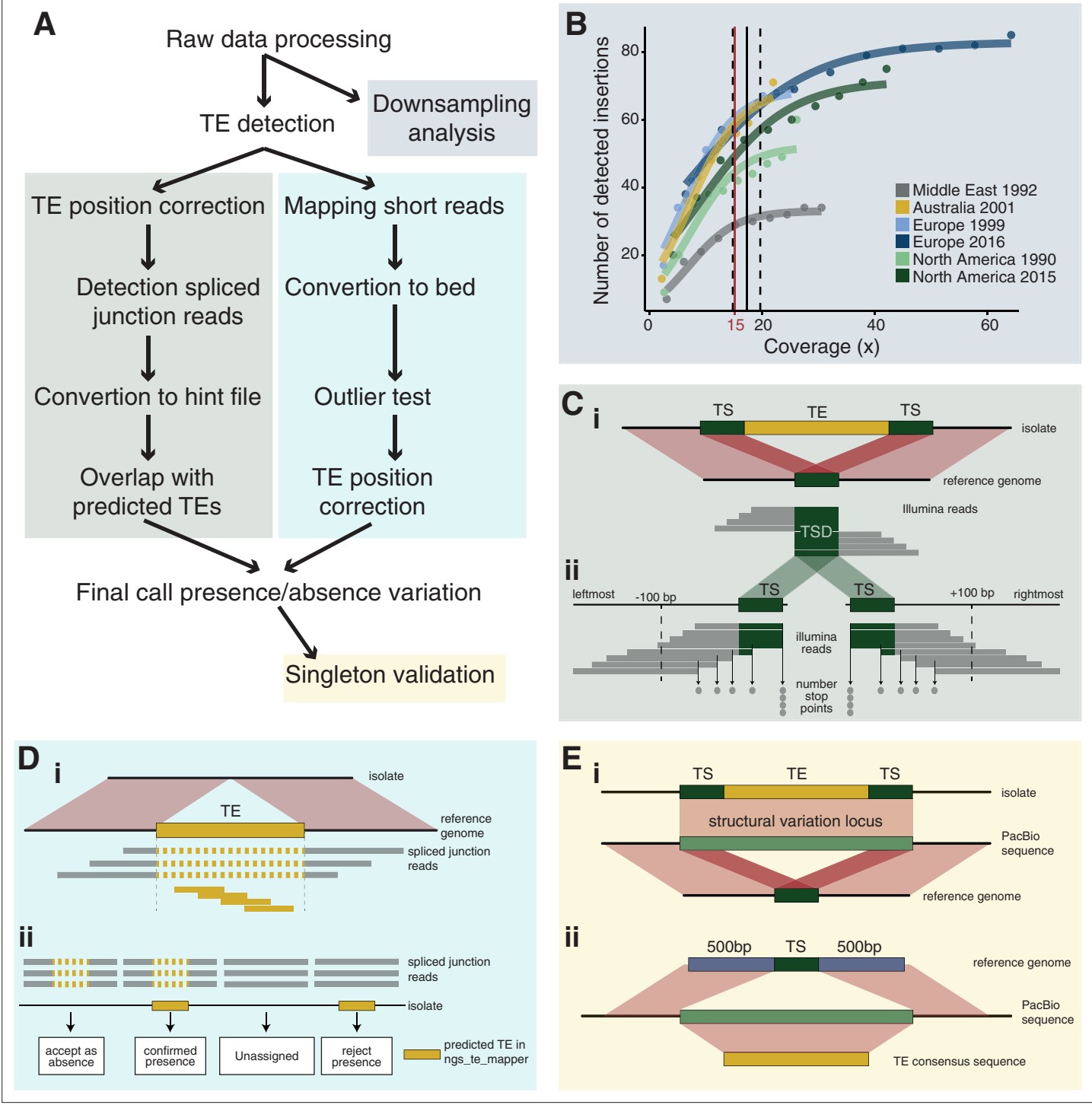

**Figure 1.** Robust discovery and validation of transposable element (TE) insertions: (**A**) General analysis pipeline. (**B**) Read depth downsampling analysis for one isolate per population with an average coverage of the population. The vertical black line indicates the coverage at which on average 90 % of the maximally detectable variants were recovered. Dashed black lines indicate the standard error. The threshold for a minimal mean coverage was set at 15 × (red line). (**C**) Validation of insertions absent in the reference genome. (**i**) TE insertions that are not present in the reference genome show a duplication of the target site and the part of the reads that covers the TE will not be mapped against the reference genome. We thus expect reads to map to the TE surrounding region and the target site duplication but not the TE itself. At the target site, a local duplication of read depth is expected. (**ii**) We selected all reads in an interval of 100 bp up- and downstream including the target site duplication to detect deviations in the number of reads terminating near the target site duplication. (**D**) Validation of insertions present in the reference genome. (**i**) Analyses read coverage at target site duplications. (**ii**) Decision map if a TE should be kept as a true insertion or rejected as a false positive. Only predicted TE insertions that overlap

*Figure 1 continued on next page*

*Figure 1 continued*

evidence of split reads were kept as TE insertions in downstream analyses. (**E**) Singleton validation using long-read PacBio sequencing. (**i**) Analysis if TE insertions overlap with a detected insertion/deletion locus (**Badet et al., 2021**). (**ii**) Homology search of the TE insertion flanking sequences based on the reference genome against PacBio reads. In addition, the consensus sequence of the inserted TE was used for matches between the flanks.

The online version of this article includes the following figure supplement(s) for figure 1:

**Source data 1.** TE insertion validations for non-reference copies.

**Source data 2.** TE consensus sequences.

**Figure supplement 1.** Validation of transposable element (TE) insertion predictions.

**Figure supplement 2.** Establishment of transposable element (TE) loci with differing start and end positions in the isolates.

**Figure supplement 3.** Bias for reads with a GC content lower than 30 % per population.

adaptive TE insertions in selective sweep regions either in the North American or European population pairs. Hence, we selected loci having low TE insertion frequencies (<5%) in all populations except either the recent North American or European population (>20%) (*Figure 3B*). Based on these criteria, we obtained seven candidate loci possibly underlying local adaptation (six in North America, one in Europe; *Figure 4A*, *Figure 4—source data 1*). All loci carry inserted retrotransposons with four RLG_Luna, one RLG_Mercurius, and one RLG_Deimos.

One TE insertion is 3815 bp downstream of a gene encoding an RTA1-like protein, which can function as transporters with a transmembrane domain and have been associated with resistance against several antifungal compounds (*Soustre et al., 1996*). The insertion is also 5785 bp upstream of a gene encoding a protein kinase domain (*Figure 4B*). The TE insertion was not detected in the Middle East or the two European populations and was at low frequencies in the Australian (3.7%) and North American 1990 (1.7%) populations, but increased to 53 % of all isolates in the North American 2015 population (fixation index $F_{ST}$ = 0.42; *Figure 4—source data 1*). Isolates that carry the insertion show a significantly higher resistance to azole antifungal compounds (*Figure 4C*). The TE is in the subtelomeric region of chromosome 12, with a moderate GC content, a low TE, and a high gene density (*Figure 4D*). The TE belongs to the family RLG_Luna, which shows a substantial burst across different chromosomes within the species (*Figure 4E,F*). We found no association between the phylogenetic relationships among isolates based on the two closest genes and the presence or absence of the TE insertion (*Figure 4G*). A second candidate adaptive TE insertion belongs to the RLG_Mercurius family and is located between two genes of unknown function (*Figure 4—figure supplement 1*). A third potentially adaptive TE insertion of a RLC_Deimos is 229 bp upstream of a gene encoding a SNARE domain protein and 286 bp upstream of a gene encoding a flavin amine oxidoreductase. Furthermore, the TE is inserted in a selective sweep region (*Figure 4—figure supplement 1*). SNARE domains play a role in vesicular transport and membrane fusion (*Bonifacino and Glick, 2004*). An additional four candidates for adaptive TE insertions belong to RLG_Luna and were located distantly to genes (*Figure 4—figure supplement 1*). We experimentally tested whether the TE insertions in proximity to genes were associated with higher levels of fungicide resistance. For this, we measured growth rates of the fungal isolates in the presence or absence of an azole fungicide widely deployed against the pathogen. We found that the insertion of TEs at two loci was positively associated with higher levels of fungicide resistance, suggesting that the adaptation was mediated by the TE (*Figure 4C*, *Figure 4—figure supplement 1*).

## Population-level expansions in TE content

If TE insertion dynamics are largely neutral across populations, TE frequencies across loci should reflect neutral population structure. To test this, we performed a principal component analysis based on a set of six populations on four continents that represent the global genetic diversity of the pathogen (*Figure 5A*) and 900,193 genome-wide SNPs (*Figure 5B*). The population structure reflected the demographic history of the pathogen with clear continental differentiation and only minor within-site differentiation. To account for the lower number of TE loci, we performed an additional principal component analysis using a random SNP set of similar size to the number of TE loci. The reduced SNP set retained the geographic signal of the broader set of SNPs (*Figure 5C*). In stark contrast, TE frequencies across loci showed only weak clustering by geographic origin with the Australian

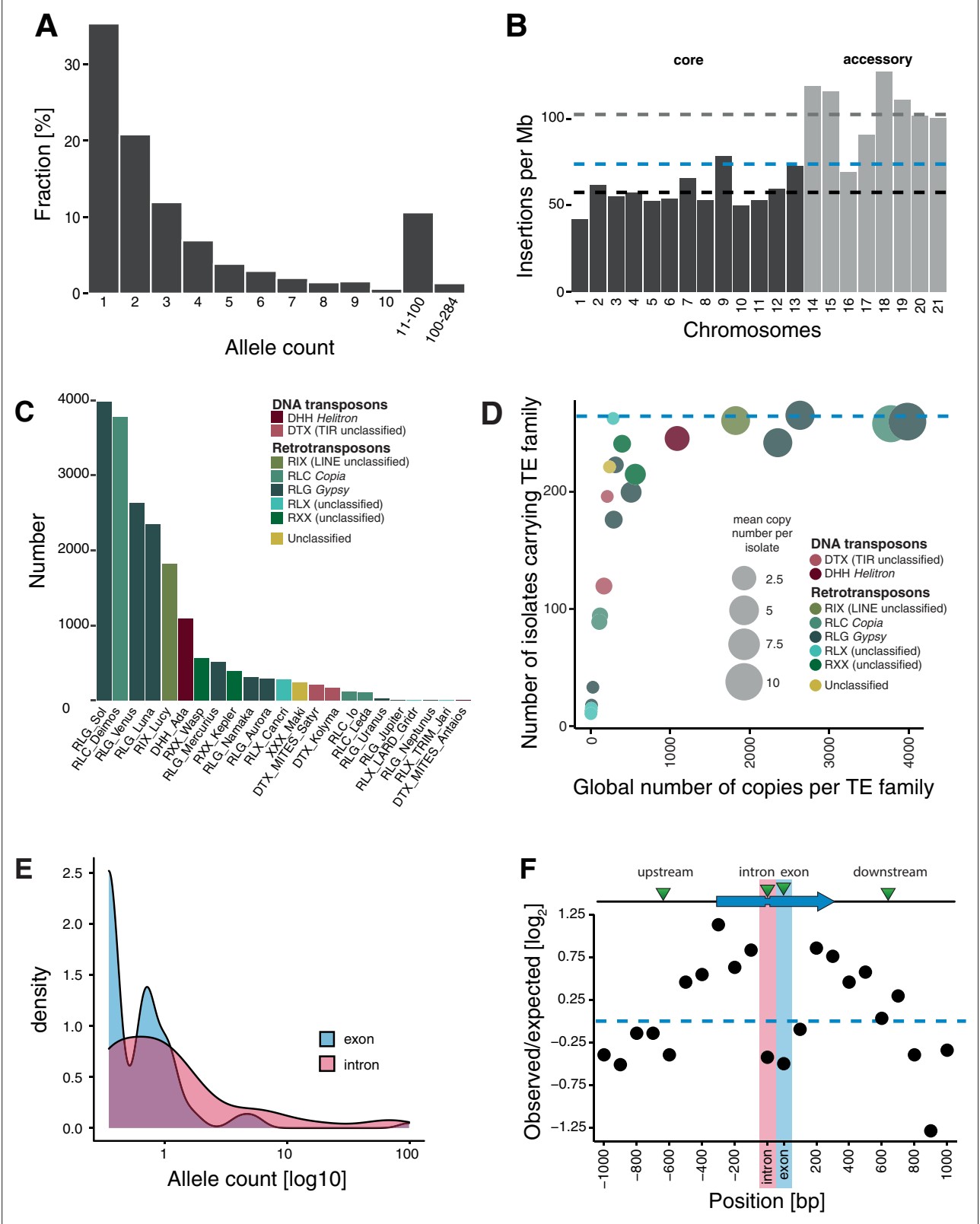

**Figure 2.** Transposable element (TE) landscape across populations. (**A**) Allele frequencies of the TE insertions across all isolates. (**B**) TE insertions per Mb on core chromosomes (dark) and accessory chromosomes (light). Dashed lines represent mean values. Blue: global mean of 75.65 insertions/Mb, dark: core chromosome mean of 58 TEs/Mb, light: accessory chromosome mean of 102.24 insertions/Mb. (**C**) Number of TE insertions per family. (**D**) TE frequencies among isolates and copy numbers across the genome. The blue line indicates the maximum number of isolates (n = 284). (**E**)

*Figure 2 continued*

Allele frequency distribution of TE insertions into introns and exons. (**F**) Number of TE insertions within 1 kb up- and downstream of genes on core chromosomes including introns and exons (100 bp windows). The blue arrow indicates a gene schematic with exons and an intron, the green triangles indicate TE insertions. The dotted blue line indicates no deviation from the expected value (i.e., mean number of TEs per window).

The online version of this article includes the following figure supplement(s) for figure 2:

**Source data 1.** TEs in reference.

**Source data 2.** Presence absence matrix TE loci.

**Source data 3.** Singletons.

**Figure supplement 1.** Validation of singleton insertions detected by mapped Illumina reads using PacBio read alignments for confirmation.

**Figure supplement 2.** TE insertion loci characteristics.

**Figure supplement 3.** Hierarchy superfamilies.

population being the most distinct (*Figure 5D*). We found a surprisingly strong differentiation of the two North American populations sampled at a 25 -year interval in the same field in Oregon.

Unusual patterns in population differentiation at TE loci suggests that TE activity may substantially vary across populations (*Figure 6*, *Figure 4—source data 1*). To analyze this, we first identified the total TE content across all loci per isolate. We found generally lower TE numbers in the Middle Eastern population from Israel (*Figure 6A–C*, *Figure 6—figure supplement 1*), which is close to the pathogen's center of origin (*Stukenbrock et al., 2007*). Populations that underwent at least one migration bottleneck showed a substantial burst of TEs across all major superfamilies. These populations included the two populations from Europe collected in 1999 and 2016 and the North American population from 1990, as well as the Australian population. We found a second stark increase in TE content in the North American population sampled in 2015 at the same site as the population from 1990. Strikingly, the isolate with the lowest number of analyzed TEs collected in 2015 was comparable to the isolate with the highest number of TEs at the same site in 1990. We tested whether sequencing coverage could explain variation in the detected TEs across isolates, but we found no meaningful association (*Figure 2—figure supplement 3C*). We analyzed whether the population-specific expansions were correlated with shifts in the frequency spectrum of TEs in the populations (*Figure 6D*). We found that the first step of expansions observed in Europe compared to the Middle East (Israel) was associated with an upwards shift in allele frequencies. This is consistent with transposition activity creating new copies in the genomes and stronger purifying selection in the Middle East. Similarly, the North American populations showed also signatures consistent with relaxation of selection against TEs (i.e., fewer low-frequency TEs). We found a significant difference (two-sample Kolmogorov–Smirnov test, two-sided) in the curve shapes between the population from the Middle East and North America 2015 (*Figure 6—source data 1*). We analyzed variation in TE copy numbers across families and found that the expansions were mostly driven by RLG elements including the families Luna, Sol, and Venus, the RLC family Deimos, and the LINE family Lucy (*Figure 6E*, *Figure 6—figure supplement 2*). We also found a North American–specific burst in DHH elements of the family Ada (increase from 4.6 to 6.1 copies on average per isolate), an increase specific to Swiss populations in LINE elements, and an increase in RLC elements in the Australian and the two North American populations. Analyses of complete *Z. tritici* reference-quality genomes that include isolates from the Israel, Australia, Switzerland (1999), and North American (1990) population revealed high TE contents in Australia and North America (Oregon 1990) (*Badet et al., 2020*). The reference-quality genomes confirmed also that the increase in TEs was driven by LINE, RLG, and RLC families in Australia and DHH, RLG, and RLC families in North America (*Badet et al., 2020*).

## TE-mediated genome size expansions

The combined effects of actively copying TE families and relaxed purifying selection lead to an accumulation of new TE insertions in populations. Consequently, mean genome sizes in populations should increase over generations. We estimated the cumulative length of TE insertions based on the length of the corresponding TE consensus sequences and found a strong increase in the total TE length in populations outside the Middle East center of origin and a second increase between the two North American populations (*Figure 7—figure supplement 1A*). To test for incipient genome expansions within the species, we first assembled genomes of all 284 isolates included in the study.

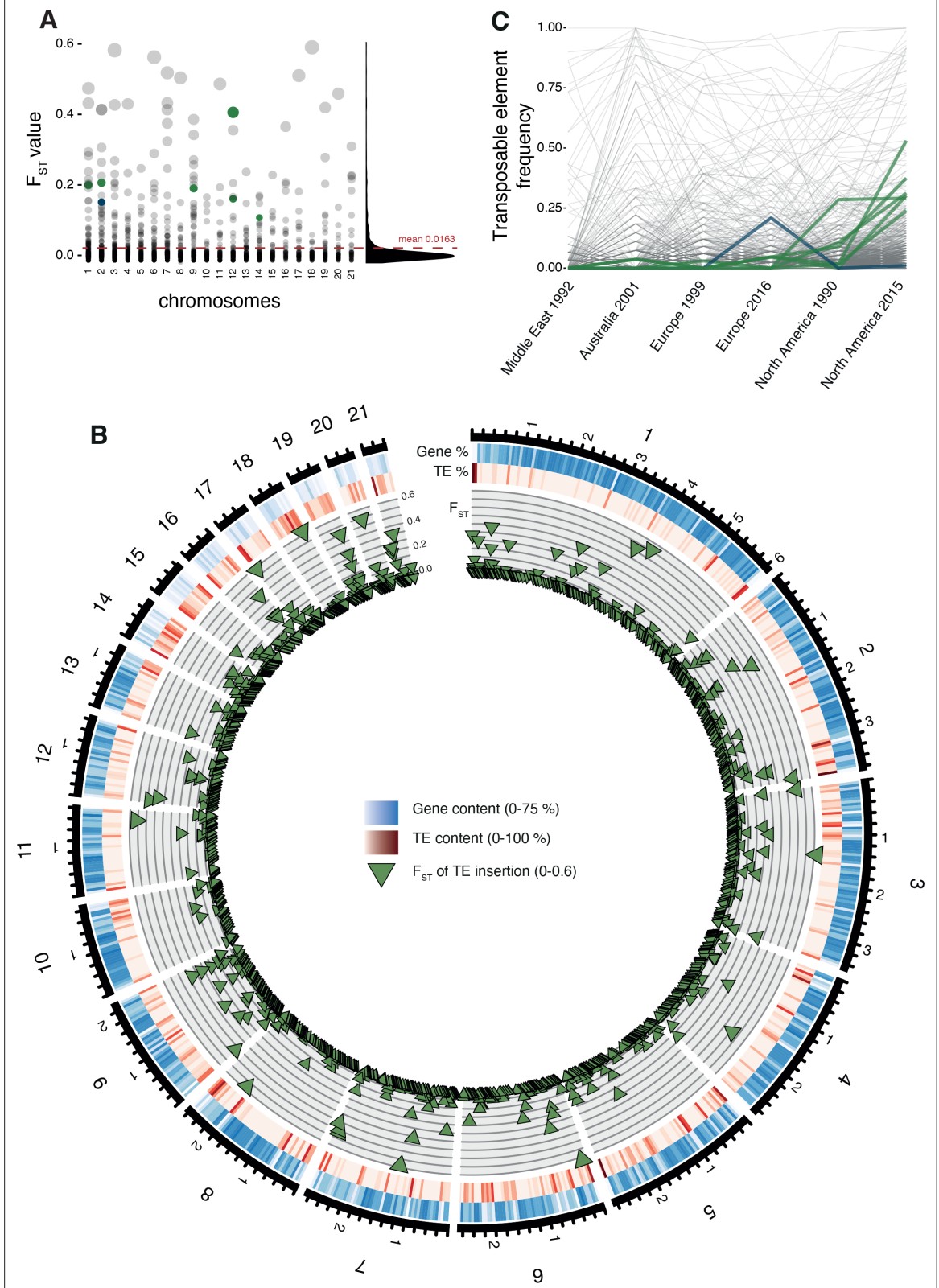

**Figure 3.** Differentiation in transposable element insertion frequencies across the genome. (**A**) Global pairwise $F_{ST}$ distributions shown across the 21 chromosomes. The red horizontal line indicates the mean $F_{ST}$ (=0.0163). TEs with a strong local short-term frequency difference among populations are highlighted (blue: increase in Europe; green: increase in North America). (**B**) Allele frequency changes between the populations. The same TE loci as in (**A**) are highlighted. (**C**) Circos plot describing from the outside to the inside: The black line indicates chromosomal position in Mb. Blue bars indicate

*Figure 3 continued on next page*

*Figure 3 continued*

the gene density in windows of 100 kb with darker blue representing higher gene density. Red bars indicate the TE density in windows of 100 kb with a darker red representing higher TE density. Green triangles indicate positions of TE insertions with among population F$_{ST}$ value shown on the y-axis.

The online version of this article includes the following figure supplement(s) for figure 3:

**Figure supplement 1.** Global pairwise FST distributions shown separately for the 21 chromosomes.

Given the limitations of short-read assemblies, we implemented corrective measures to compensate for potential variation in assembly qualities. We corrected for variation in the GC content of different sequencing datasets by downsampling reads to generate balanced sequencing read sets prior to assembly (see Materials and methods). We also excluded all reads mapping to accessory chromosomes because different isolates are known to differ in the number of these chromosomes. Genome assemblies were checked for completeness by retrieving the phylogenetically conserved BUSCO genes (*Figure 7A*). Genome assemblies across different populations carry generally >99% complete BUSCO gene sets, matching the completeness of reference-quality genomes of the same species (*Badet et al., 2020*). The completeness of the assemblies showed no correlation with either TE or GC content of the genomes. GC content was inversely correlated with genome size consistent with the expansion of repetitive regions having generally low GC content (*Figure 7B*). We found that the core genome size varied substantially among populations with the Middle East, Australia, as well as the two older European and North American populations having the smallest core genome sizes (*Figure 7C*). We found a notable increase in core genome size in both the more recent European and North American populations. The increase in core genome size is positively correlated with the count and cumulative length of all inserted TEs (*Figure 7D, E and G*) and negatively correlated with the genome-wide GC content (*Figure 7F,G*, *Benjamini and Speed, 2012*). Hence, core genome size shows substantial variation within the species matching the recent expansion in TEs across continents. We found the most variable genome sizes in the more recent North American population (*Figure 7—figure supplement 1B*). Finally, we contrasted variation in genome size with the detected TE insertion dynamics. For this, we assessed the variable genome segment as the difference between the smallest and largest analyzed core genome. To reflect TE dynamics, we calculated the cumulative length of all detected TE insertions in any given genome. We found that the cumulative length of inserted TEs represents between 4.8% and 184 % of the variable genome segment defined for the species or 0.2–2.6% of the estimated genome size per isolate (*Figure 7—figure supplement 1C,D*).

## Discussion

TEs play a crucial role in generating adaptive genetic variation within species but are also drivers of deleterious genome expansions. We analyzed the interplay of TEs with selective and neutral processes including population differentiation and incipient genome expansions. TEs have substantial transposition activity in the genome but are strongly counterselected and are maintained at low frequency. TE dynamics showed distinct trajectories across populations with more recently established populations having higher TE content and a concurrent expansion of the genome.

### Recent selection acting on TE insertions

TE frequencies in the species show a strong skew toward singleton insertions across populations. However, our short read based analyses are possibly skewed toward over-counting singletons as indicated by independent long-read mapping evaluations. Nevertheless, the skew toward low-frequency TE insertions indicates both that TEs are undergoing transposition and that purifying selection maintains frequencies at a low level. Similar effects of selection on active TEs were observed across plants and animals, including *Drosophila melanogaster* and *Brachypodium distachyon* (*Cridland et al., 2013*; *Stritt et al., 2018*; *Luo et al., 2020*). TE insertions were under-represented in or near coding regions, showing a stronger purifying selection against TEs inserting into genes. Coding sequences in the *Z. tritici* genome are densely packed with an average distance of only ~1 kb (*Goodwin et al., 2011*). Consistent with this high gene density, TE insertions were most frequent at a distance of 200–400 bp away from coding sequences. A rapid decay in linkage disequilibrium in the *Z. tritici* populations (*Croll et al., 2015*; *Hartmann et al., 2018*) likely contributed to the efficiency of removing deleterious insertions. Some TE superfamilies have preferred insertion sites in coding regions and transcription start

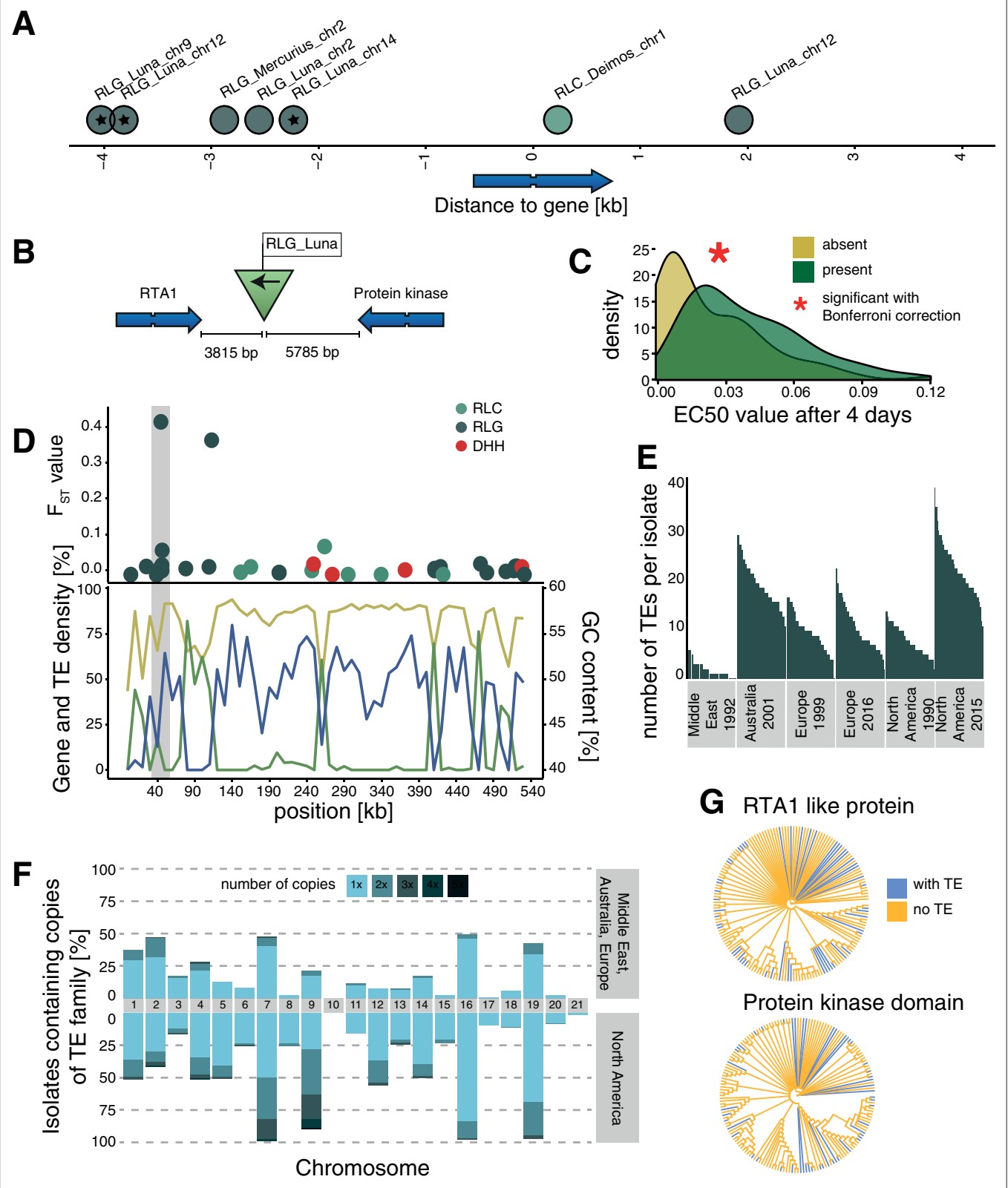

**Figure 4.** Candidate adaptive transposable element (TE) insertions. (**A**) Distribution of all extremely differentiated TEs and their distance to the closest gene. Color indicates the superfamily. The stars indicate TE insertions not found in the reference genome. (**B**) Location of the RLG_Luna TE insertion on chromosome 12 corresponding to its two closest genes. (**C**) Resistance against azole fungicides among isolates as a function of TE presence or absence. (**D**) Genomic niche of the RLG_Luna TE insertion on chromosome 12: $F_{ST}$ values for each TE insertion, gene content (blue), TE content (green) and GC

*Figure 4 continued on next page*

*Figure 4 continued*

content (yellow). The gray section highlights the insertion site. (**E**) Number of RLG_Luna copies per isolate and population. (**F**) Frequency changes of RLG_Luna between the two North American populations compared to the other populations. Colors indicate the number of copies per chromosome. (**G**) Phylogenetic trees of the coding sequences of either the gene encoding the RTA1-like protein or the protein kinase domain. Isolates of the two North American populations and an additional 11 isolates from other populations not carrying the insertion are shown. Blue color indicates TE presence, yellow indicates TE absence.

The online version of this article includes the following figure supplement(s) for figure 4:

**Source data 1.** Top loci information.

**Figure supplement 1.** Additional top loci.

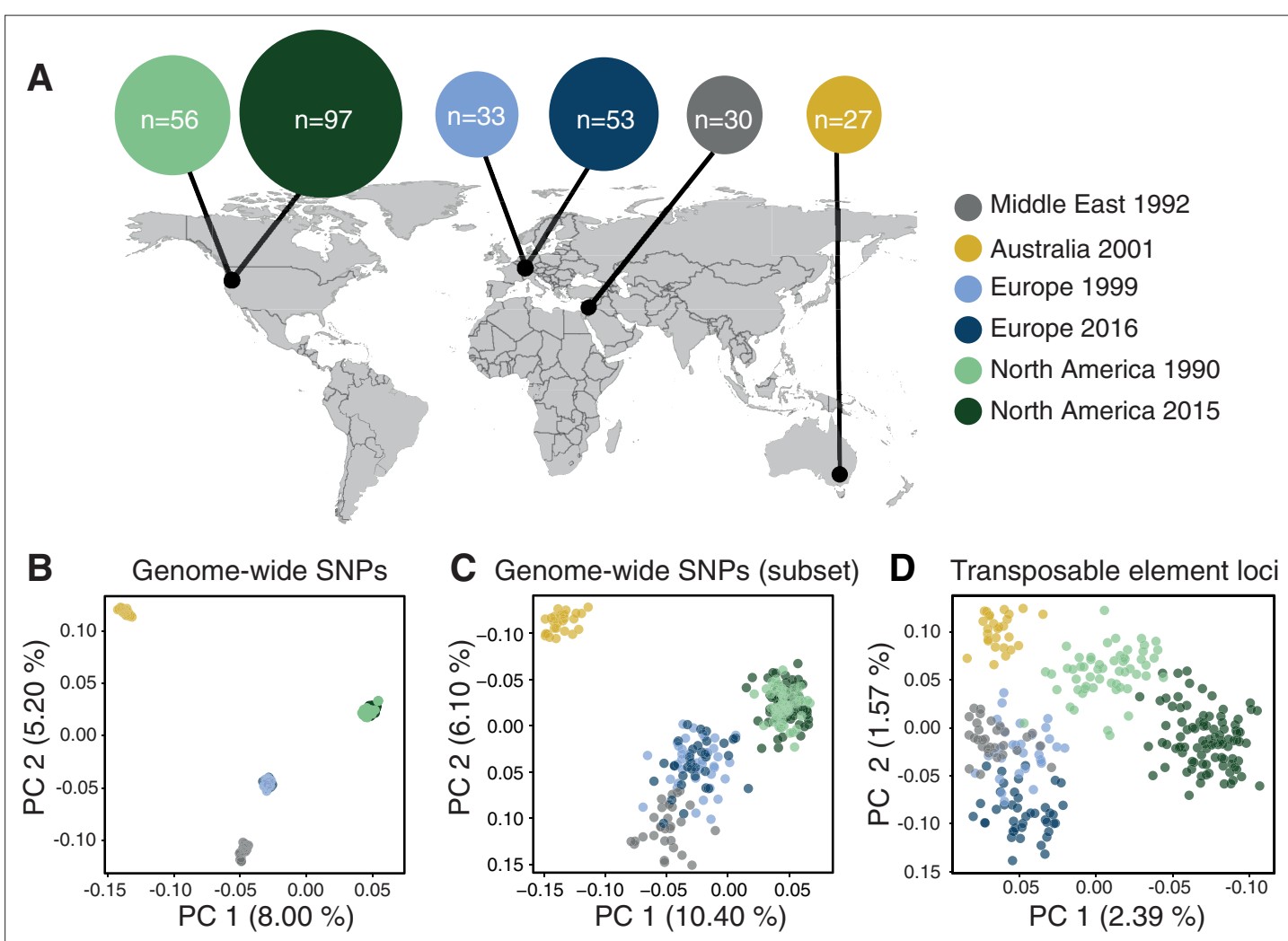

**Figure 5.** Population differentiation at transposable element (TE) and genome-wide SNP loci. (**A**) Sampling locations of the six populations. Middle East represents the region of origin of the pathogen. In North America, the two populations were collected at an interval of 25 years in the same field in Oregon. In Europe, two populations were collected at an interval of 17 years from two fields in Switzerland <20 km apart. Dark arrows indicate the historic colonization routes of the pathogen. (**B**) Principal component analysis (PCA) of 284 *Zymoseptoria tritici* isolates, based on 900,193 genome-wide SNPs. (**C**) PCA of a reduced SNP data set with randomly selected 203 SNPs matching approximately the number of analyzed TE loci. (**D**) PCA based on 193 TE insertion loci. Loci with allele frequency <5 % are excluded.

The online version of this article includes the following figure supplement(s) for figure 5:

**Source data 1.** Isolates.

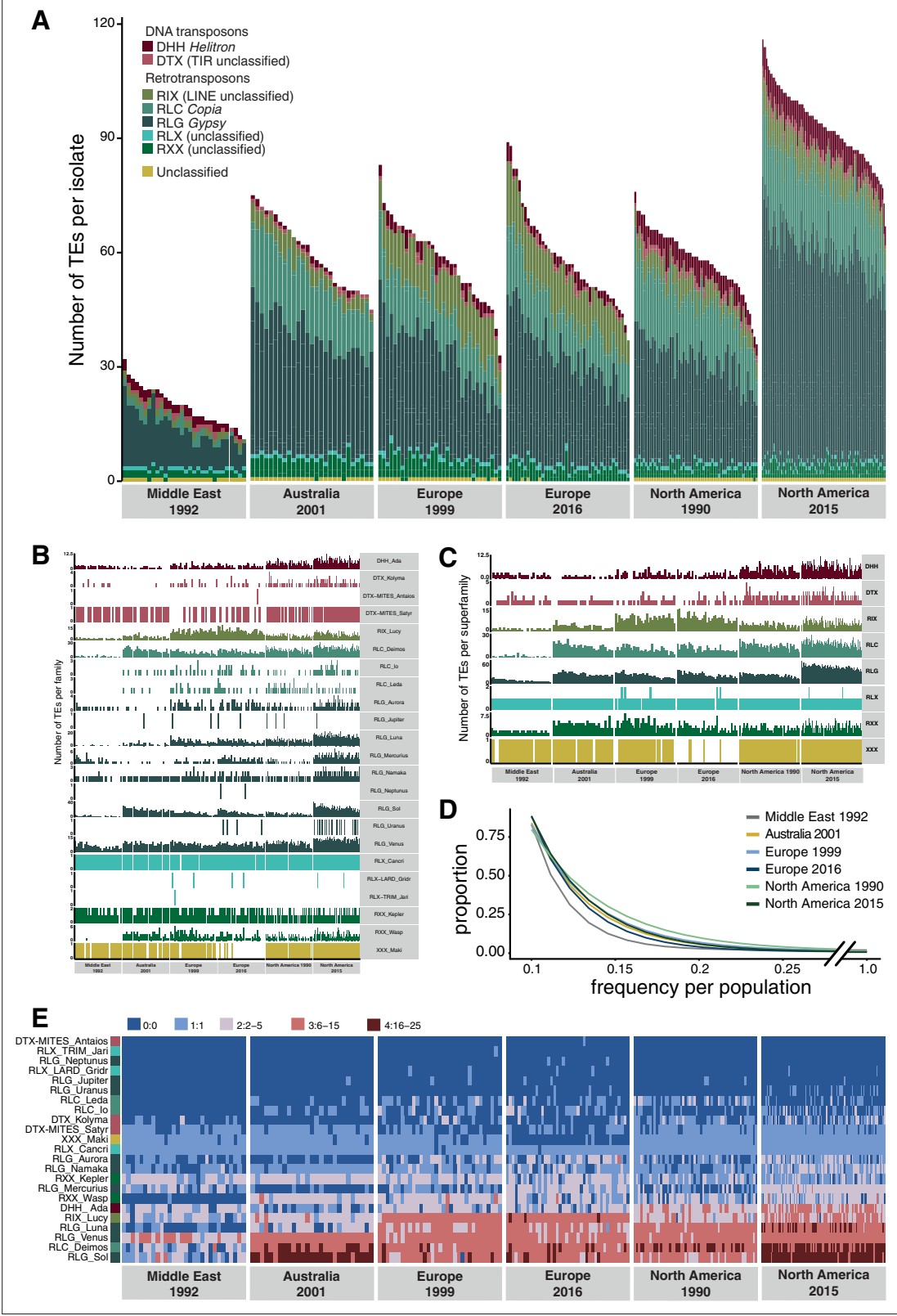

**Figure 6.** Global population structure of transposable element (TE) insertion polymorphism. (**A**) Total TE copies per isolate. Colors identify TE superfamilies. (**B**) TE copies per family and (**C**) superfamily. (**D**) TE insertion frequency spectrum per population. The curve fitting was performed with a self-starting Nls asymptomatic regression model (**E**). TE family copy numbers per isolate.

*Figure 6 continued on next page*

*Figure 6 continued*

The online version of this article includes the following figure supplement(s) for figure 6:

**Source data 1.** Kolmogorof–Smirnov.

**Figure supplement 1.** Population changes additional.

**Figure supplement 2.** Heatmap loci.

sites (*Miyao et al., 2003*; *Fu et al., 2013*; *Gilly et al., 2014*; *Quadrana et al., 2016*). Hence, some heterogeneity in the observed insertion site distribution across the genome is likely due to insertion preferences of individual TEs. We also found evidence for positive selection acting on TEs with the strongest candidate locus being a TE insertion on chromosome 12. This locus showed a frequency increase only in the more recent North American population, which experienced the first systematic fungicide applications and subsequent emergence of fungicide resistance in the decade prior to the last sampling (*Estep et al., 2015*). The nearest gene encodes a RTA1-like protein, a transmembrane exporter that is associated with resistance toward different stressors, including antifungal compounds, and shows strong copy number variation in several fungi (*Soustre et al., 1996*; *Rogers and Barker, 2003*; *Sirisattha et al., 2004*; *Ali et al., 2013*; *Yew et al., 2016*; *Liang et al., 2018*). Hence, the TE insertion may have positively modulated RTA1 expression to resist antifungals.

Transposition activity in a genome and counteracting purifying selection are expected to establish an equilibrium over evolutionary time (*Charlesworth and Charlesworth, 2009*). However, temporal bursts of TE families and changes in population size due to bottlenecks or founder events are likely to shift the equilibrium. Despite purifying selection, we were able to detect signatures of positive selection by scanning for short-term population frequency shifts. Population genomic datasets can be used to identify the most likely candidate loci underlying recent adaptation. The shallow genome-wide differentiation of *Z. tritici* populations provides a powerful background to test for outlier loci (*Hartmann et al., 2018*). We found the same TE families to have experienced genome-wide copy number expansions, suggesting that the availability of adaptive TE insertions may be a by-product of TE bursts in individual populations.

## Population-level TE invasions and relaxed selection

Across the surveyed populations from four continents, we identified substantial variation in TE counts per genome. The increase in TEs matches the global colonization history of the pathogen with an increase in TE copies in more recently established populations (*Zhan et al., 2003*; *Stukenbrock et al., 2007*). Compared to the Israeli population located nearest the center of origin in the Middle East, the European populations showed a threefold increase in TE counts. The Australian and North American populations established from European descendants retained high TE counts. We identified a second increase at the North American site where TE counts nearly doubled again over a 25 year period. Compared to the broader increase in TEs from the Middle East, the second expansion at the North American site was driven by a small subset of TE families alone. Analyses of completely assembled reference-quality genomes from the same populations confirmed that genome expansions were primarily driven by the same TE families belonging to the RLG, RLC, and DHH superfamilies (*Badet et al., 2020*). Consistent with the contributions from individual TEs, we found that the first expansion in Europe led to an increase in low-frequency variants, suggesting higher transposition activity of many TEs in conjunction with strong purifying selection. The second expansion at the North American site shifted TE frequencies upwards, suggesting relaxed selection against TEs. The population-level context of TEs in *Z. tritici* shows how heterogeneity in TE control interacts with demography to determine extant levels of TE content and, ultimately, genome size.

## TE invasion dynamics underpins genome size expansions

The number of detected TEs was closely correlated with core genome size; hence, genome size expansions were at least partly caused by the very recent proliferation of TEs. Genome assemblies of large eukaryotic genomes based on short-read sequencing are often fragmented and contain chimeric sequences (*Nagarajan and Pop, 2013*). Focusing on the less repetitive core chromosomes in the genome of *Z. tritici* reduces such artifacts substantially. Because genome assemblies are the least complete in the most repetitive regions, any under-represented sequences may rather underestimate

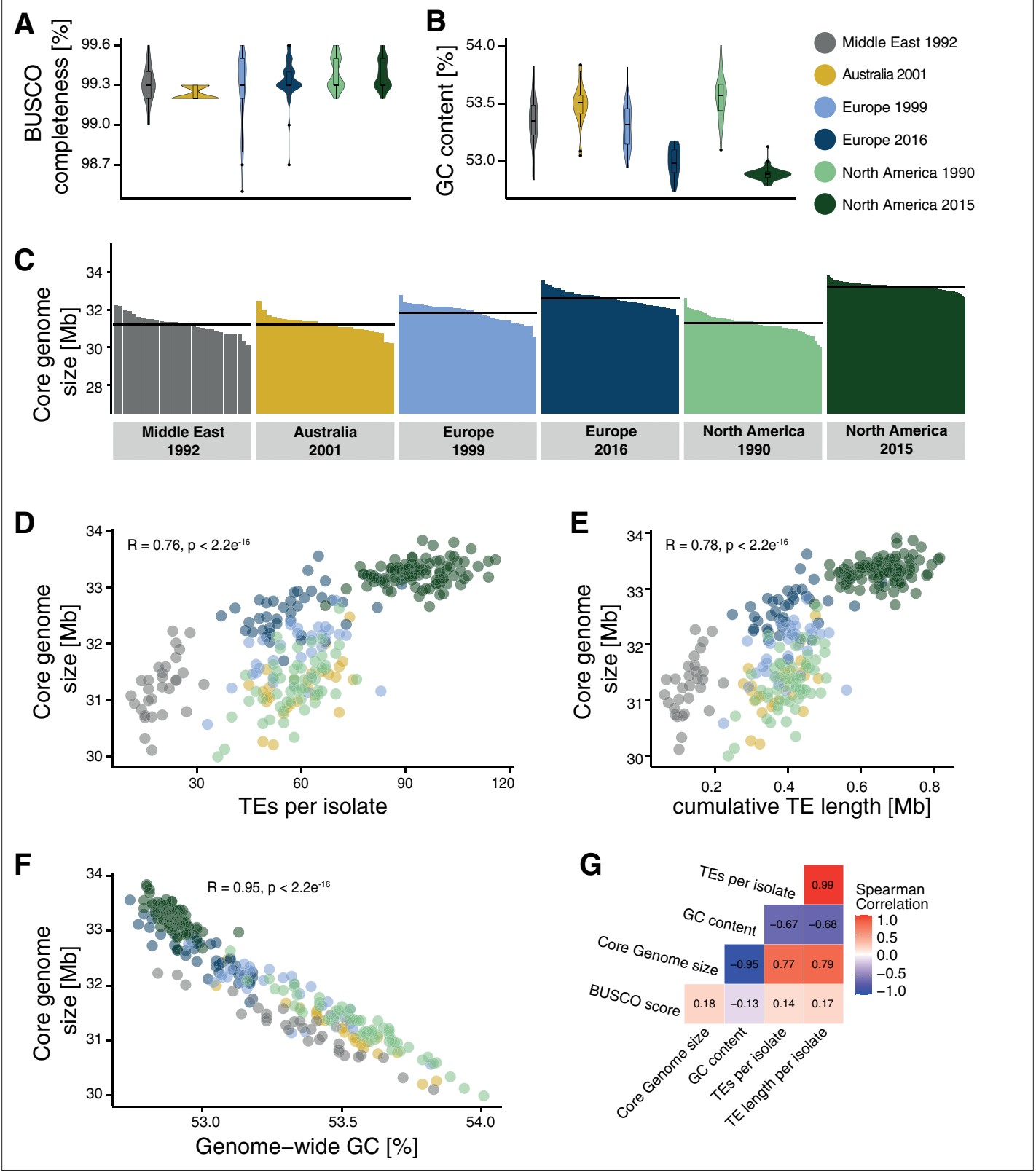

**Figure 7.** Core genome size and transposable element (TE) evolution across populations. (**A**) BUSCO completeness variation among genome assemblies. Black lines indicate the mean genome size per population. (**B**) Genome-wide GC content variation. (**C**) Core genome size variation among the isolates of the populations (excluding accessory chromosomes). (**D**) Correlation of core genome size and number of detected TEs. (**E**) Correlation of core genome size and the cumulative length of all TEs detected as inserted. (**F**) Correlation of core genome size and genome-wide GC content. (**G**)

*Figure 7 continued*

Spearman correlation matrix of BUSCO completeness, core genome size, number of detected TEs, and genome-wide GC content.

The online version of this article includes the following figure supplement(s) for figure 7:

**Figure supplement 1.** Genome size expansion.

than overestimate within-species variation in genome size. Hence, we consider the assembly sizes to be a robust correlate of total genome size. The core genome size differences observed across the species range match genome size variation typically observed among closely related species. Among primates, genome size varies by ~70 % with ~10 % between humans and chimpanzees (*Rogers and Gibbs, 2014*; *Miga et al., 2020*). In fungi, genome size varies by several orders of magnitude within phyla but is often highly similar among closely related species (*Raffaele and Kamoun, 2012*). Interestingly, drastic changes in genome size have been observed in the *Blumeria* and *Pseudocercospora* genera where genome size changed by 35–130% between the closest known species (*González-Sayer et al., 2021*; *Frantzeskakis et al., 2018*). Beyond analyses of TE content variation correlating with genome size evolution, proximate mechanisms driving genome expansions are poorly understood. By establishing large population genetic datasets, such as those possible for crop pathogens, analyses of genome size evolution become tractable at the population level.

TEs might not only contribute to genome expansion directly by adding length through additional copies, but also by increasing the rate of chromosomal rearrangements and ectopic recombination (*Bourque et al., 2018*; *Blommaert, 2020*). However, TEs are not the only repetitive elements that can lead to a genome size expansion. In *Arabidopsis thaliana* genomes, the 45 S rDNA has been shown to have the strongest impact on genome size variation, followed by 5 S rDNA variation, and contributions by centromeric repeats and TEs (*Long et al., 2013*). In conjunction, recent work demonstrates how repetitive sequences are drivers of genome size evolution over short evolutionary timescales.

The activity of TEs is controlled by complex selection regimes within species. Actively transposing elements may accelerate genome evolution and underpin expansions. Hence, genomic defenses should evolve to efficiently target recently active TEs. Here, we show that TE activity and counteracting genomic defenses have established a tenuous equilibrium across the species range. We show that population subdivisions are at the origin of highly differentiated TE content within a species matching genome size changes emerging over the span of only decades and centuries. In conclusion, population-level analyses of genome size can recapitulate genome expansions typically observed across much deeper time scales providing fundamentally new insights into genome evolution.

## Materials and methods
### Fungal isolate collection and sequencing
We analyzed 295 *Z. tritici* isolates covering six populations originating from four geographic locations and four continents (*Figure 1—source data 1*), including: Middle East 1992 (n = 30 isolates, Nahal Oz, Israel), Australia 2001 (n = 27, Wagga Wagga), Europe 1999 (n = 33, Berg am Irchel, Switzerland), Europe 2016 (n = 52, Eschikon, ca. 15 km from Berg am Irchel, Switzerland), and North America 1990 and 2015 (n = 56 and n = 97, Willamette Valley, Oregon; *McDonald et al., 1996*; *Linde et al., 2002*; *Zhan et al., 2002*; *Zhan et al., 2003*; *Zhan et al., 2005*). Illumina short-read data from the Middle Eastern, Australian, European 1999, and North American 1990 populations were obtained from the NCBI Sequence Read Archive (SRA) under the BioProject PRJNA327615 (*Hartmann et al., 2017*). For the Switzerland 2016 and Oregon 2015 populations, asexual spores were harvested from infected wheat leaves from naturally infected fields and grown in YSB liquid media including 50 mgL$^{-1}$ kanamycin and stored in silica gel at −80 °C. High-quality genomic DNA was extracted from liquid cultures using the DNeasy Plant Mini Kit from Qiagen (Venlo, The Netherlands). The isolates were sequenced on an Illumina HiSeq in paired-end mode and raw reads were deposited at the NCBI SRA under the BioProject PRJNA596434.

### TE insertion detection
The quality of Illumina short reads was determined with FastQC version 0.11.5 (https://www.bioinformatics.babraham.ac.uk/projects/fastqc/) (*Figure 1A*). To remove spuriously sequenced Illumina

adaptors and low-quality reads, we trimmed the sequences with Trimmomatic version 0.36, using the following filter parameters: illuminaclip:TruSeq3-PE-2.fa:2:30:10 leading:10 trailing:10 sliding-window:5:10 minlen:50 (*Bolger et al., 2014*). We created repeat consensus sequences for TE families (sequences are available on https://github.com/crolllab/datasets (copy archived at swh:1:rev:364745 6a2f7ed986b690501d690f09d02d3f586e); *Laboratory of Evolutionary Genetics @ UNINE, 2021*; *Figure 1—source data 2*) in the complete reference genome IPO323 (*Goodwin et al., 2011*) with RepeatModeler version open-4.0.7 (http://www.repeatmasker.org/RepeatModeler/) based on the RepBase Sequence Database and de novo (*Bao et al., 2015*). TE classification into superfamilies and families was based on an approach combining detection of conserved protein sequences and tools to detect non-autonomous TEs (*Badet et al., 2020*). To detect TE insertions, we used the R-based tool ngs_te_mapper version 79ef861f1d52cdd08eb2d51f145223fad0b2363c integrated into the McClintock pipeline version 20cb912497394fabddcdaa175402adacf5130bd1, using bwa version 0.7.4-r385 to map Illumina short reads, samtools version 0.1.19 to convert alignment file formats and R version 3.2.3 (*Li and Durbin, 2009*; *Li et al., 2009*; *Linheiro and Bergman, 2012*; *R Development Core Team, 2017*; *Nelson et al., 2017*).

## Downsampling analysis

We performed a downsampling analysis to estimate the sensitivity of the TE detection with ngs_te_mapper based on variation in read depth. We selected one isolate per population matching the average coverage of the population. We extracted the per-base pair read depth with the genomecov function of bedtools version 2.27.1 and calculated the genome-wide mean read depth (*Quinlan and Hall, 2010*). The number of reads in the original fastq file was reduced in steps of 10 % to simulate the impact of reduced coverage. We analyzed each of the obtained reduced read subsets with ngs_te_mapper using the same parameters as described above. The correlation between the number of detected insertions and the read depth was visualized using the function nls with model SSlogis in R and visualized with ggplot2 (*Wickham, 2016*). The number of detected TEs increased with the number of reads until reaching a plateau indicating saturation (*Figure 1B*). Saturation was reached at a coverage of approximately 15 ×; hence, we retained only isolates with an average read depth above 15 × for further analyses. We thus excluded one isolate from the Oregon 2015 population and 10 isolates from the Switzerland 2016 population.

## Validation procedure for predicted TE insertions

ngs_te_mapper detects the presence but not the absence of a TE at any given locus. We devised additional validation steps to ascertain both the presence and the absence of a TE across all loci in all individuals. TEs absent in the reference genome were validated by re-analyzing mapped Illumina reads. Reads spanning both parts of a TE sequence and an adjacent chromosomal sequence should only map to the reference genome sequence and cover the target site duplication of the TE (*Figure 1C*). We used bowtie2 version 2.3.0 with the parameter `--very-sensitive-local` to map Illumina short reads of each isolate on the reference genome IPO323 (*Langmead and Salzberg, 2012*). Mapped Illumina short reads were then sorted and indexed with samtools, and the resulting bam file was converted to a bed file with the function bamtobed in bedtools. We extracted all mapped reads with an end point located within 100 bp of the target site duplication (*Figure 1C*). We tested whether the number of reads with a mapped end around the target site duplication significantly deviated if the mapping ended exactly at the boundary. A mapped read ending exactly at the target site duplication boundary is indicative of a split read mapping to a TE sequence absent in the reference genome. To test for the deviation in the number of read mappings around the target site duplication, we used a Poisson distribution and the *ppois* function in R version 3.5.1 (*Figure 1C*). We identified a TE as present in an isolate if tests on either side of the target site duplication had a p-value<0.001 (*Figure 5—source data 1*; *Figure 1—figure supplement 1B* and *Figure 1—source data 1*).

For TEs present in the reference genome, we analyzed evidence for spliced junction reads spanning the region containing the TE. Spliced reads are indicative of a discontinuous sequence and, hence, absence of the TE in a particular isolate (*Figure 1D*). We used STAR version 2.5.3a to detect spliced junction reads with the following set of parameters: `--runThreadN` 1 `--outFilterMultimapNmax` 100 `--winAnchorMultimapNmax` 200 `--outSAMmultNmax` 100 `--outSAMtype` BAM Unsorted `--outFilterMismatchNmax` 5 `--alignIntronMin` 150 `--alignIntronMax` 15,000

(*Dobin et al., 2013*). We then sorted and indexed the resulting bam file with samtools and converted split junction reads with the function bam2hints in bamtools version 2.5.1 (*Barnett et al., 2011*). We selected loci without overlapping spliced junction reads using the function intersect in bedtools with the parameter `-loj -v`. We considered a TE as truly absent in an isolate if `ngs_te_mapper` did not detect a TE and evidence for spliced junction reads were found, indicating that the isolate had no inserted TE in this region. If the absence of a TE could not be confirmed by spliced junction reads, we labeled the genotype as missing. Finally, we excluded TE loci with more than 20 % missing data from further investigations (*Figure 1D*, *Figure 1—figure supplement 1C*).

## Clustering of TE insertions into loci

We identified insertions across isolates as being the same locus if all detected TEs belonged to the same TE family and insertion sites differed by ≤100 bp (*Figure 1—figure supplement 2*). We used the R package *GenomicRanges* version 1.28.6 with the functions makeGRangesFromDataFrame and findOverlaps and the R package *devtools* version 1.13.4 (*Lawrence et al., 2013*; *Wickham and Chang, 2016*). We used the R package *dplyr* version 0.7.4 to summarize datasets (https://dplyr.tidyverse.org/). Population-specific frequencies of insertions were calculated with the function allele.count in the R package *hierfstat* version 0.4.22 (*Goudet, 2005*). We conducted a principal component analysis for TE insertion frequencies filtering for a minor allele frequency ≥5 %. We also performed a principal component analysis for genome-wide single nucleotide polymorphism (SNP) data obtained from *Hartmann et al., 2017* and *Singh et al., 2020*. As described previously, SNPs were hard-filtered with VariantFiltration and SelectVariants tools integrated in the Genome Analysis Toolkit (GATK) (*McKenna et al., 2010*). SNPs were removed if any of the following filter conditions applied: QUAL < 250; QD < 20.0; MQ < 30.0; –2 > BaseQRankSum > 2; –2 > MQRankSum > 2; –2 > ReadPosRankSum > 2; FS > 0.1. SNPs were excluded with vcftools version 0.1.17 and plink version 1.9 requiring a genotyping rate > 90% and a minor allele frequency > 5% (https://www.cog-genomics.org/plink2, *Chang et al., 2015*). Finally, we converted tri-allelic SNPs to bi-allelic SNPs by recoding the least frequent allele as a missing genotype. Principal component analysis was performed using the *gdsfmt* and *SNPRelate* packages in R (*Zheng et al., 2012*; *Zheng et al., 2017*). For a second principal component analysis with a reduced set of random markers, we randomly selected SNPs with vcftools and the following set of parameters: `--maf` 0.05 –thin 200,000 to obtain an approximately equivalent number of SNPs as TE loci.

## Evaluation of singleton insertions

To evaluate the reliability of singleton TE insertion loci, we analyzed singleton loci in isolates for which we had both Illumina datasets and complete reference-quality genomes (*Badet et al., 2020*). From a set of 19 long-read PacBio reference genomes spanning the global distribution of *Z. tritici*, one isolate each from Australia, Israel, North America (1990) and four isolates from Europe (1999) were also included in the TE insertion screening. To assess the reliability of singleton TE insertions, we first investigated structural variation analyses among the reference genomes (*Badet et al., 2021*: Supplementary Data 1 and 2). The structural variation was called both based on split read mapping of PacBio reads and pairwise whole-genome alignments. Using bedtools intersect, we recovered for the 31 singleton TE loci in the seven analyzed genomes a total of 17 loci showing either an indel, translocation, copy number polymorphism, duplication, inverted duplication, inversion, or inverted translocation at the same location. We visually inspected the PacBio read alignment bam files against the IPO323 reference genome using IGV version 2.4.16 (*Robinson et al., 2011*), and found a typical coverage increase at the target site duplication, with most read mappings interrupted at the target site duplication as expected for an inserted TE. For the 14 remaining TE loci, we extracted the region of the predicted insertion and padded the sequence on both ends with an additional 500 bp using samtools faidx. We used blast to identify a homologous region in the assembled reference-quality genomes. Matching regions were inspected based on blastn for the presence of a TE sequence matching the TE family originally detected at the locus. With this second approach, we confirmed an additional five singletons to be true insertions. Both methods combined produced supportive evidence for 22 of 31 singleton insertions (71%). We calculated the read coverage after mapping to the reference genome IPO323 with bedtools genomecov for each PacBio long-read dataset and calculated mean coverage for 500 bp regions around singleton TE insertions.

## Population differentiation in TE frequencies

We calculated Nei's fixation index ($F_{ST}$) between pairs of populations using the R packages *hierfstat* and *adegenet* version 2.1.0 (*Jombart, 2008*; *Jombart and Ahmed, 2011*). To understand the chromosomal context of TE insertion loci across isolates, we analyzed draft genome assemblies. We generated de novo genome assemblies for all isolates using SPAdes version 3.5.0 with the parameter `--careful` and a kmer range of '21, 29, 37, 45, 53, 61, 79, 87' (*Bankevich et al., 2012*). We used blastn to locate genes adjacent to TE insertion loci on genomic scaffolds of each isolate. We then extracted scaffold sequences surrounding 10 kb up- and downstream of the localized gene with the function faidx in samtools and reverse complemented the sequence if needed. Then, we performed multiple sequence alignments for each locus across all isolates with MAFFT version 7.407 with parameter `--maxiterate` 1000 (*Katoh and Standley, 2013*). We performed visual inspections to ensure correct alignments across isolates using Jalview version 2.10.5 (*Waterhouse et al., 2009*). To generate phylogenetic trees of individual gene or TE loci, we extracted specific sections of the alignment using the function extractalign in EMBOSS version 6.6.0 (*Rice et al., 2000*) and converted the multiple sequence alignment into PHYLIP format with jmodeltest version 2.1.10 using the -getPhylip parameter. We then estimated maximum likelihood phylogenetic trees with the software PhyML version 3.0, the K80 substitution model, and 100 bootstraps on the ATGC South of France bioinformatics platform (*Guindon and Gascuel, 2003*; *Guindon et al., 2010*; *Darriba et al., 2012*). Bifurcations with a supporting value lower than 10 % were collapsed in TreeGraph version 2.15.0–887 beta, and trees were visualized as circular phylograms in Dendroscope version 2.7.4 (*Huson et al., 2007*; *Stöver and Müller, 2010*). For loci showing complex rearrangements, we generated synteny plots using 19 completely sequenced genomes from the same species using the R package *genoplotR* version 0.8.9 (*Guy et al., 2010*; *Badet et al., 2020*). We calculated the population-specific allele frequency for TE loci and estimated the exponential decay curve with a self-starting Nls asymptomatic regression model nls(p_loci ~ SSasymp(p_round, Asym, R0, lrc)) in R.

We analyzed signatures of selective sweeps based on genome-wide SNPs using the extended haplotype homozygosity (EHH) tests implemented in the R package *REHH* (*Sabeti et al., 2007*; *Gautier and Vitalis, 2012*). We analyzed within-population signatures based on the iHS statistic and chose a maximum gap distance of 20 kb. We also analyzed cross-population signatures based on the XP-EHH statistic for the following two population pairs: North America 1990 versus North America 2015, Europe 1999 versus Europe 2016. We defined significant selective sweeps as being among the 99.9th percentile outliers of the iHS and XP-EHH statistics. Significant SNPs at less than 5 kb were clustered into a single selective sweep region adding ±2.5 kb. Finally, we analyzed whether TE loci in the population pairs were within 10 kb of a region identified as a selective sweep by XP-EHH using the function intersect from bedtools.

## Genomic location of TE insertions

To characterize the genomic environment of TE insertion loci, we split the reference genome into non-overlapping windows of 10 kb using the function splitter from EMBOSS. TEs were located in the reference genome using RepeatMasker providing consensus sequences from RepeatModeler (http://www.repeatmasker.org/). To analyze coding sequence, we retrieved the gene annotation for the reference genome (*Grandaubert et al., 2015*). We estimated the percentage covered by genes or TEs per window using the function intersect in bedtools. Additionally, we calculated the GC content using the tool get_gc_content (https://github.com/spundhir/RNA-Seq/blob/master/get_gc_content.pl; *Sachin, 2021*). We extracted the number of TEs present in 1 kb windows around each annotated core gene in the reference genome IPO323, using the function window in bedtools. We calculated the relative distances between each gene and the closest TE with the function bedtools closest. For the TEs inserted into genes, we used the function intersect in bedtools to distinguish intron and exon insertions with the parameters -wo and -v, respectively. TEs that overlap more than one exon were only counted once. For each 100 bp segment in the 1 kb windows as well as for introns and exons, we calculated the mean number of observed TE insertions per base pair. We calculated the mean number of TEs per window and calculated the log2 of the observed number of TE insertions divided by the expected value. We extracted information about recombination hotspots from *Croll et al., 2015*. This dataset is based on two experimental crosses initiated from isolates included in our analyses (1A5 × 1E4, 3D1 × 3D7). The recombination rates were assessed based on the reference genome IPO323

and analyzed with the *R/qtl* package in R. We used bedtools intersect to compare both TE density in IPO323 and TE insertion polymorphism with predicted recombination hotspots.

## Core genome size estimation

Accessory chromosomes show the presence/absence variation within the species and length polymorphism (*Goodwin et al., 2011*; *Croll et al., 2013*) and thus impact genome size. We controlled for this effect by first mapping sequencing reads to the reference genome IPO323 using bowtie2 with `--very-sensitive-local` settings and retained only reads mapping to any of the 13 core chromosomes using seqtk subseq v1.3-r106 (https://github.com/lh3/seqtk/; *Heng, 2021*). Furthermore, we found that different sequencing runs showed minor variation in the distribution of the per read GC content. In particular, reads of a GC content lower than 30 % were under-represented in the Australian (mean reads <30 % of the total readset: 0.05%), North American 1990 (0.07%) and Middle East (0.1%) populations, and higher in the Europe 1999 (1.3%), North American 2015 (3.0%), and Europe 2016 (4.02%) populations (*Figure 1—figure supplement 3*). Library preparation protocols and Illumina sequencer generations are known factors influencing the recovery of reads of varying GC content (*Benjamini and Speed, 2012*).

To control a potential bias stemming from this, we subsampled reads based on GC content to create homogeneous datasets. For this, we first retrieved the mean GC content for each read pair using geecee in EMBOSS and binned reads according to GC content. For the bins with a GC content <30%, we calculated the mean proportion of reads from the genome over all samples. We then used seqtk subseq to subsample reads of <30% to adjust the mean GC content among readsets. We generated de novo genome assemblies using the SPAdes assembler version with the parameters `--careful` and a kmer range of '21, 29, 37, 45, 53, 61, 79, 87'. The SPAdes assembler is optimized for the assembly of relatively small eukaryotic genomes. We evaluated the completeness of the assemblies using BUSCO v4.1.1 with the fungi_odb10 gene test set (*Simão et al., 2015*). We finally ran Quast v5.0.2 to retrieve assembly metrics including scaffolds of at least 1 kb (*Mikheenko et al., 2018*).

## Fungicide resistance assay

To quantify susceptibility toward propiconazole, we used a previously published microtiter plate assay dataset with three replicates performed for each isolate and concentration. Optical density was used to estimate growth rates under different fungicide concentrations (0, 0.00006, 0.00017, 0.0051, 0.0086, 0.015, 0.025, 0.042, 0.072, 0.20, 0.55, 1.5 mg $L^{-1}$) (*Hartmann et al., 2020*). We calculated dose–response curves and estimated the half-maximal lethal concentration $EC_{50}$ with a four-parameter logistics curve in the R package *drc* (*Ritz and Streibig, 2005*).

# Acknowledgements

We thank Andrea Sánchez Vallet, Anne C Roulin, Luzia Stalder, Adam Taranto, Emilie Chanclud, and Alice Feurtey for helpful discussions and comments on previous versions of the manuscript. We also thank the three reviewers for very helpful suggestions. We thank C Sarai Reyes-Avila for advice on statistical analyses. DC is supported by the Swiss National Science (grants 31,003A_173265) and the Fondation Pierre Mercier pour la Science.

# Additional information

## Funding

| Funder | Grant reference number | Author |
| --- | --- | --- |
| Schweizerischer Nationalfonds zur Förderung der Wissenschaftlichen Forschung | 31003A_173265 | Daniel Croll |
| Pierre Mercier pour la science | | Daniel Croll |

| Funder | Grant reference number | Author |
|--------|------------------------|--------|

The funders had no role in study design, data collection and interpretation, or the decision to submit the work for publication.

## Author contributions

Ursula Oggenfuss, Conceptualization, Data curation, Formal analysis, Investigation, Methodology, Writing - original draft; Thomas Badet, Formal analysis; Thomas Wicker, Data curation, Supervision; Fanny E Hartmann, Data curation, Resources, Supervision; Nikhil Kumar Singh, Leen Abraham, Petteri Karisto, Christopher Mundt, Bruce A McDonald, Resources; Tiziana Vonlanthen, Investigation; Daniel Croll, Conceptualization, Funding acquisition, Methodology, Project administration, Resources, Supervision, Writing - original draft, Writing - review and editing

## Author ORCIDs

Ursula Oggenfuss http://orcid.org/0000-0001-9291-9185
Thomas Badet http://orcid.org/0000-0001-6130-441X
Fanny E Hartmann http://orcid.org/0000-0002-9365-4008
Nikhil Kumar Singh http://orcid.org/0000-0001-7236-9278
Petteri Karisto http://orcid.org/0000-0003-4807-0190
Christopher Mundt http://orcid.org/0000-0002-1216-7583
Bruce A McDonald http://orcid.org/0000-0002-5332-2172
Daniel Croll http://orcid.org/0000-0002-2072-380X

## Decision letter and Author response

Decision letter https://doi.org/10.7554/eLife.69249.sa1
Author response https://doi.org/10.7554/eLife.69249.sa2

## Additional files

### Supplementary files

• Transparent reporting form

### Data availability

Sequence data are deposited at the NCBI Sequence Read Archive under the accession numbers PRJNA327615, PRJNA596434 and PRJNA178194. Transposable element consensus sequences are available from https://github.com/crolllab/datasets (copy archived at https://archive.softwareheritage.org/swh:1:rev:3647456a2f7ed986b690501d690f09d02d3f586e).

The following dataset was generated:

| Author(s) | Year | Dataset title | Dataset URL | Database and Identifier |
|-----------|------|---------------|-------------|-------------------------|
| Hartmann FE, McDonald BA, Croll D | 2019 | Population sequencing of Zymoseptoria tritici in Switzerland and Oregon (USA) | https://www.ncbi.nlm.nih.gov/bioproject/PRJNA596434 | NCBI BioProject, PRJNA596434 |

The following previously published datasets were used:

| Author(s) | Year | Dataset title | Dataset URL | Database and Identifier |
|-----------|------|---------------|-------------|-------------------------|
| Hartmann FE, McDonald BA, Croll D | 2016 | Population genomics of Zymoseptoria tritici | https://www.ncbi.nlm.nih.gov/bioproject/PRJNA327615/ | NCBI BioProject, PRJNA327615 |
| Croll D, McDonald BA | 2012 | Zymoseptoria tritici genome sequencing | https://www.ncbi.nlm.nih.gov/bioproject/PRJNA178194/ | NCBI BioProject, PRJNA178194 |

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
