## [Decision Letter]

**Acceptance summary:**

This study will appeal to a broad audience in evolutionary and population genomics. The work reveals how demographic processes have shaped transposable element dynamics in an important fungal pathogen. Many of the conclusions are likely to apply to other taxa as well.

**Decision letter after peer review:**

Thank you for submitting your article "A population-level invasion by transposable elements triggers genome expansion in a fungal pathogen" for consideration by *eLife*. Your article has been reviewed by 3 peer reviewers, and the evaluation has been overseen by Detlef Weigel as the Senior and Reviewing Editor. The following individuals involved in review of your submission have agreed to reveal their identity: Marie Mirouze (Reviewer #1); Zoé Joly-Lopez (Reviewer #2); Leandro Quadrana (Reviewer #3).

Essential revisions:

1) Since calling TE insertion polymorphisms is an inherently difficult business, we strongly encourage you to validate a select number of polymorphic TE insertions using either a long read dataset or conventional molecular biology, focusing on singleton TEs, i.e., those present only in one strain. The reviewers provided several additional suggestions to exclude artifacts as much as possible.

2) Link TE density and insertion polymorphisms to local recombination rates.

3) Add statistical analyses to support qualitative statements.

4) There was confusion about the sweep scans. Please spell out in the text that TEs were associated with sweeps after these had been identified based on SNPs (at least this is my understanding of the methods; if this is wrong, please consider including sweep scans with SNPs and indels). Along these lines, it might be worth saying something about using TE insertion polymorphisms directly in sweep scans.

5) Address the apparent discrepancies between figure 4 and supplementary figure 12 and the attendant conclusions.

The reviewers made a series of additional excellent suggestions, which we ask you to pay attention to.

*Reviewer #1 (Recommendations for the authors):*

I enjoyed reading the paper. Here are a few points that could help clarifying your findings:

– It would be nice to validate some of the TIPS using either classical PCR or long read data.

– Most families have relatively low copy numbers (Figure S6). Is the increase in copy number sufficient to explain the genome size increase (given that a TE might be generally smaller than 10 kb on average)?

– It could be interesting to highlight what figures are based on core chromosomes and what figures include all chromosomes, as for genomicists not familiar with fungal genomes this distinction would not be evident. This is stated on line 306 but could be explicit for each figure.

– Line 283: The authors have already published the complete genomes, could they be more specific on the difference between the two datasets? Is the 2020 paper focused on one "reference" genome per geographic origin?

– Line 352: Is MFS1 corresponding to the MFS gene on Figure 4B? If so, what is the MSF1 expression level in the US genomes (or in the TE-present versus TE-absent individuals ?)– Figure 1D ii is not clear to me.– Figures S6 and S12 are very interesting and could be moved to main figures (if no space constraint).

– It would be interesting to see Figure 3a with the color-code for RLG vs RLC and DHH TE classes.

– It seems that the 5 selected candidate adaptive TEs (line 204) are not comprised in the 5 ones found in selective sweep regions (line 190). If so, could the 5 ones on selective sweep regions be cited in the text maybe?

–An example of TE burst within Oryza could be cited (Piegu et al., 2006, see Line 74).

*Reviewer #2 (Recommendations for the authors):*

While the type of dataset that the authors have can lead to interesting hypothesis testing for TE dynamics, the manuscript lacked clarity between the hypothesis to be tested, the results, and the link to the figures, which makes it unfortunately difficult to measure the impact of the findings.

1. The fungicide assay is used to test for potential TE insertions conferring fungicide resistance. The authors do not mention which insertion loci are being tested and the legends in the supplementary Figure 12 does not provide details. Counting 26 loci helped to then make the probable link with the 26 candidate loci with local adaptation in the North American populations, but then the link with the five TE insertion in proximity to genes (line 204) is not clear. Are the 3 significant loci part of the five described in Figure 4 and supplemental Figures 8-11? How do the authors reconcile these results with the fact that the Helitron insertion highlighted in Figure 4 is not significant in the fungicide assay (by trying to read the insertion names in Supp. Figure S12). Finally, the Methods section lacks key details as to the number of replicates for the assay and statistical analyses.

2.The authors looked at the frequency spectrum of TEs in the 6 populations to see if they correlated with potential population-specific expansions. This would provide a good indicator about the type of selection acting on these populations. The panel 6D does not support the conclusions. It is hard to understand the bar graph and it is unclear to the reader how the authors are using the frequency bins to conclude the shift. Based on the changes of the color bars, the results for the North American population do not seem to be supported. Unless there was an issue with the labeling of the figure, or that additional information in the legends should be provided to make the results stand out.

3. I may not have understood, but the authors found five loci overlapping selective sweep regions and mention that they are retrotransposons. Then the story changes its focus on 5 insertions loci in proximity to genes associated with fungicide resistance or host adaptation. It takes a good part of the paragraph to understand which insertion refers to the ones overlapping the sweep. I would have liked a more detailed description of these selective sweep region, especially regarding the regions not being associated with the fungicide resistance genes.

4. The authors mention migration bottleneck, which I assume they refer to population bottleneck caused by migration. Would it be possible to provide references to support this assumption?

5. I would suggest that the authors carefully review the figures and the corresponding legends. For example:

– Figure 2. Modify order so that the panels are called in the proper order. In A, green triangle not defined, 2C, hard to see the red vs grey line, 2E hard to distinguish among the colors, like RIX and RLG gypsy and provide legends for all dot sizes, 2F: no hexagon, do we mean arrow?

– Figure 2F: not clear where the singleton are indicated or not, but called for in the manuscript. What does "more than two copies in the populations" mean? I would assume it is two or more (not more than two) but it is not clear whether it is more than two copies per population or at least one copy in multiple populations.

– Figure 3. Dots present next to some of the chromosome numbers and it is not explained in the legends why.

– Line 179: Was it intended to say European Population or really the Australian? If so, justify why.

– Some of the Figure panels are not called in the main text (only in methods). For example only Figure 1A is mentioned in the text. I would like to see the panels B, C, D being integrated in the text to clarify how the pipeline works and the validation.

– Figure 4G in its current state does not support the conclusion that it is only found in North American populations as only the Middle-East population is compared, which was probably chosen since it represents the origin of the pathogen.

*Reviewer #3 (Recommendations for the authors):*

Line 166-171. I do not fully understand the analysis of the data presented in Figure 2A. It is not clear from the results and methods section how the expected distribution of TE insertions was obtained. In addition, it seems that TE insertions located very close of two genes may be counted twice, introducing potential biases in the analysis (i.e. insertions in gene-rich and gene-poor regions are counted twice or once, respectively). Independently of how this analysis was performed, I disagree with your interpretation of the results based only on selective forces, as the localization of TE insertions is the results of integration preferences and subsequent selective biases. For instance, it has been established that COPIA and MuDR transposons integrate preferentially within or upstream genes in multiple species. The effect of integration preference is particularly important in this article, as most TE insertion polymorphisms detected singletons that are expected to be the result of very recent transposition events unfiltered by natural selection.

Line 205-213 The increased frequency of the helitron insertion at MFS could be the consequence of either positive selection of the TE insertion, a linked polymorphism (genetic hitchhiking) or else background selection on the haplotype. Better evidence for positive selection on the TE insertion itself could be obtained by scanning the region for selective sweeps based on SNPs and polymorphic TEs and using site frequency spectrum-based (E.g Fst) and/or haplotype length-based (e.g. iHS). Selective sweeps signatures centered on the TE insertion would support selection on this variant.

Line 225-227. The association between TE presence and azole resistance is tantalizing, but cannot imply causality. As pointed before, the TE insertions may be simply hitchhiking a causal mutation. Moreover, I cannot find in the article any result supporting that the TE insertions associated with azole resistance also show signatures of positive selection. Indeed, it seems that none of the insertion discussed at length in Lines 205-223 show associations in Suppl. Figure 12. This is a key point of the article because without this connection it is difficult to draw any conclusion about potential benefits associated with TE insertions in Zymoseptoria tritici.

Line 354-357. As I mentioned above, I don't see any correspondence between signatures of positive selection and azole resistance for the Gypsy (RLG_Sol) insertion near the MFS gene in Chr 12 (Figure S10). Indeed, this insertion is not marked as statistically significant in Figure S11.

Figure 6D. The reduction observed in the fraction of loci with low allele frequency (10%) in North American isolates is not accompanied by an increase in the proportion of loci at high allele frequency. In other words, it seems as if the bars do not sum up 1. Please check/clarify this figure.

Work in Arabidopsis demonstrated that intraspecific genome size variation is largely explained by copy number variation in 45S rRNA (Long et al. 2013 Nat. Genet). This work should be discussed in relation to your observations, particularly given that the de novo assembly of genomes using short-reads is unable to estimate copy number variation of rRNA or centromeres.

---

## [Author Response]

Essential revisions:1) Since calling TE insertion polymorphisms is an inherently difficult business, we strongly encourage you to validate a select number of polymorphic TE insertions using either a long read dataset or conventional molecular biology, focusing on singleton TEs, i.e., those present only in one strain. The reviewers provided several additional suggestions to exclude artifacts as much as possible.

We entirely agree that singleton calls in particular can benefit from additional validation (note that our procedure had already multiple validation steps beyond the standard ngs_te_mapper pipeline). Following the recommendations below, we have now taken advantage of PacBio long-read datasets to specifically screen for singleton candidates. We had in total access to seven PacBio high-coverage sequencing sets overlapping with isolates included in the previous version of the manuscript. We now report the outcome of the seven isolates in detail in the manuscript (methods/results/suppl. figures/tables). In short, we could confirm ~70% of singletons (or 22 out of 31 singletons) in all tested isolates combined. A more thorough explanation of our approach and findings is also provided below.

2) Link TE density and insertion polymorphisms to local recombination rates.

This is an excellent suggestion. We do have access to high-resolution recombination maps for the species and we integrated this now in our analyses. Overall, we found that the TE density in the reference genome (that includes also older and nested insertions) shows no difference between genome-wide and recombination hotspots. We did detect a trend towards higher TE numbers outside of recombination hotspots (adjusted for the breadth of hotspots). We mention this now in the Results and added additional panels to figure supplements.

3) Add statistical analyses to support qualitative statements.

We have carefully checked our statements and added statistical significance testing where appropriate.

4) There was confusion about the sweep scans. Please spell out in the text that TEs were associated with sweeps after these had been identified based on SNPs (at least this is my understanding of the methods; if this is wrong, please consider including sweep scans with SNPs and indels). Along these lines, it might be worth saying something about using TE insertion polymorphisms directly in sweep scans.

We now much more clearly spell out that the selection scans were based on genome-wide SNP sets (not directly TE loci). As requested, we also explain in more detail what genes / TE insertion loci were found in sweep regions. We also explain now that including TE insertion loci directly into the scans maybe problematic due to the uneven coverage and heterogeneity in linkage disequilibrium. We do think though that the population differentiation scans of TE insert frequencies provide relevant information about candidate TEs involved in adaptation.

5) Address the apparent discrepancies between figure 4 and supplementary figure 12 and the attendant conclusions.

We now more carefully explain the contents of these two figures. We replaced the candidate adaptive TE locus to increase coherence. According to a comment below, we added parts of the figure supplement to Figure 4.

The reviewers made a series of additional excellent suggestions, which we ask you to pay attention to.Reviewer #1 (Recommendations for the authors):I enjoyed reading the paper. Here are a few points that could help clarifying your findings:– It would be nice to validate some of the TIPS using either classical PCR or long read data.

We entirely agree. Following up from the response above, we have used PacBio long-read data generated on an overlapping set of isolates to validate specifically singleton TEs. These are potentially the most problematic ones. In a subset of seven isolates for which we also have high-coverage PacBio long read sequences, we could confirm ~70% of singleton TEs. We describe this now in methods/results and supplementaries. Furthermore, we caution in the discussion how errors in singleton detection may affect a specific subset of our findings.

– Most families have relatively low copy numbers (Figure S6). Is the increase in copy number sufficient to explain the genome size increase (given that a TE might be generally smaller than 10 kb on average)?

This is an excellent suggestion. We have now estimated the expected sequence length added by the inserted TEs in populations with enlarged genomes and added Figure 7E to compare increased genome size and cumulative length of TEs. We added cumulative TE length to the Spearman correlation matrix as well. We find that these sequences explain ~0.19-2.55% of the genome size increase. We added an additional figure supplement to show the estimation of cumulated TE length and amount of explanation of TE length on genome size expansion. It is important to note that we present only a subset of the totality of the TE insertions due to the fact that we surely are unable to capture all TE insertions with our approach. The stringent quality filtering additionally contributes to this underrepresentation. We think that TE-mediated rearrangements and duplications could additionally explain the gap between the reported TE increases and the estimated genome size increase. We discuss these aspects now more thoroughly.

– It could be interesting to highlight what figures are based on core chromosomes and what figures include all chromosomes, as for genomicists not familiar with fungal genomes this distinction would not be evident. This is stated on line 306 but could be explicit for each figure.

We make this now more explicit as suggested by renaming “genome size” to “core genome size”.

– Line 283: The authors have already published the complete genomes, could they be more specific on the difference between the two datasets? Is the 2020 paper focused on one "reference" genome per geographic origin?

We now more explicitly explain the geographic breadth of each dataset when we mention the pangenome publication. Yes, the 2020 publication established single reference genomes (total n=19) for different regions without accounting for within-region diversity.

– Line 352: Is MFS1 corresponding to the MFS gene on Figure 4B? If so, what is the MSF1 expression level in the US genomes (or in the TE-present versus TE-absent individuals?)

Made now clearer.

– Figure 1D ii is not clear to me.

We modified the figure and improved the legend.

– Figures S6 and S12 are very interesting and could be moved to main figures (if no space constraint).

We are happy to do so. We now present S6 as part of main figure 6 and parts of S12 as part of main figure 4.

– It would be interesting to see Figure 3a with the color-code for RLG vs RLC and DHH TE classes.

We now provide a new figure supplement using a clearer color code to distinguish these specific superfamilies.

– It seems that the 5 selected candidate adaptive TEs (line 204) are not comprised in the 5 ones found in selective sweep regions (line 190). If so, could the 5 ones on selective sweep regions be cited in the text maybe?

We now provide the missing information. We ran the selective sweep analysis on iHS and XP-EHH separately. We changed the filtering order: (a) first filter for all TEs that are located in a region of selective sweep, (b) filter elements with high allele frequency increase in US/Oregon or Europe (c) then order by global pairwise F_ST_. This led to us discussing a different candidate in the main texts.

–An example of TE burst within Oryza could be cited (Piegu et al., 2006, see Line 74).

Thank you. Citation added.

Reviewer #2 (Recommendations for the authors):While the type of dataset that the authors have can lead to interesting hypothesis testing for TE dynamics, the manuscript lacked clarity between the hypothesis to be tested, the results, and the link to the figures, which makes it unfortunately difficult to measure the impact of the findings.1. The fungicide assay is used to test for potential TE insertions conferring fungicide resistance. The authors do not mention which insertion loci are being tested and the legends in the supplementary Figure 12 does not provide details. Counting 26 loci helped to then make the probable link with the 26 candidate loci with local adaptation in the North American populations, but then the link with the five TE insertion in proximity to genes (line 204) is not clear. Are the 3 significant loci part of the five described in Figure 4 and supplemental Figures 8-11?

We apologize for the lack of clarity here. We now make the links between the different observation more explicit. We specifically mention how the datasets presented in the different figures are linked. According to reviewer #1 suggestions we moved the supplementary figure content to main Figure 5.

How do the authors reconcile these results with the fact that the Helitron insertion highlighted in Figure 4 is not significant in the fungicide assay (by trying to read the insertion names in Supp. Figure S12).

We agree that there is not enough clarity to link the different methods to find candidate TEs. We changed the filtering methods for candidates with a stronger focus onTE in regions of selective sweep (see above).

Finally, the Methods section lacks key details as to the number of replicates for the assay and statistical analyses.

We have now improved the methods section with the missing information.

2.The authors looked at the frequency spectrum of TEs in the 6 populations to see if they correlated with potential population-specific expansions. This would provide a good indicator about the type of selection acting on these populations. The panel 6D does not support the conclusions. It is hard to understand the bar graph and it is unclear to the reader how the authors are using the frequency bins to conclude the shift. Based on the changes of the color bars, the results for the North American population do not seem to be supported. Unless there was an issue with the labeling of the figure, or that additional information in the legends should be provided to make the results stand out.

We are grateful for these suggestions. We have now improved the visualization in the panel (now 6D) using fitted curves and pairwise comparison with Kolmogorov-Smirnov. We also make the legend more complete and discuss more coherently what the differences in frequency spectra suggests.

3. I may not have understood, but the authors found five loci overlapping selective sweep regions and mention that they are retrotransposons. Then the story changes its focus on 5 insertions loci in proximity to genes associated with fungicide resistance or host adaptation. It takes a good part of the paragraph to understand which insertion refers to the ones overlapping the sweep. I would have liked a more detailed description of these selective sweep region, especially regarding the regions not being associated with the fungicide resistance genes.

We have now supplemented the Results with a summary of what was found in the selective sweep regions. We also more clearly refer to how different TE insertions overlap with selective sweep regions.

4. The authors mention migration bottleneck, which I assume they refer to population bottleneck caused by migration. Would it be possible to provide references to support this assumption?

This was indeed insufficiently explained. We now refer to previous studies on global colonization history, recent gene flow and population-level diversity. We then explain more clearly how some populations have undergone a population bottleneck.

5. I would suggest that the authors carefully review the figures and the corresponding legends. For example:– Figure 2. Modify order so that the panels are called in the proper order. In A, green triangle not defined, 2C, hard to see the red vs grey line, 2E hard to distinguish among the colors, like RIX and RLG gypsy and provide legends for all dot sizes, 2F: no hexagon, do we mean arrow?

Errors corrected, graphical improvements made as suggested and other figures are reviewed as well.

– Figure 2F: not clear where the singleton are indicated or not, but called for in the manuscript. What does "more than two copies in the populations" mean? I would assume it is two or more (not more than two) but it is not clear whether it is more than two copies per population or at least one copy in multiple populations.

We agree with the reviewer that this figure is not too clear. In the original figure singletons and TE loci with only two occurrences were not included. We changed the figure to a density plot and clarified the wording in the text and legend.

– Figure 3. Dots present next to some of the chromosome numbers and it is not explained in the legends why.

We removed the dots. We initially intended to help distinguish the numbers 6 and 9.

– Line 179 : Was it intended to say European Population or really the Australian ? If so, justify why.

We thank the reviewers for this remark and changed the text accordingly.

– Some of the Figure panels are not called in the main text (only in methods). For example only Figure 1A is mentioned in the text. I would like to see the panels B,C,D being integrated in the text to clarify how the pipeline works and the validation.

We have now checked all figures/panels, so that we mention each sequentially in the Results. Figure 1B-D were mentioned in the methods before.

– Figure 4G in its current state does not support the conclusion that it is only found in North American populations as only the Middle-East population is compared, which was probably chosen since it represents the origin of the pathogen.

We added more reference genomes to the synteny plot. While DHH_Ada is not present in this location in any other isolate, the Australian isolate shows at a similar locus a high density of several nested TE insertions

Reviewer #3 (Recommendations for the authors):Line 166-171. I do not fully understand the analysis of the data presented in Figure 2A. It is not clear from the results and methods section how the expected distribution of TE insertions was obtained. In addition, it seems that TE insertions located very close of two genes may be counted twice, introducing potential biases in the analysis (i.e. insertions in gene-rich and gene-poor regions are counted twice or once, respectively).

We have now clarified in the legend and methods how we avoid double-counting and biases. This is now Figure 2F.

Independently of how this analysis was performed, I disagree with your interpretation of the results based only on selective forces, as the localization of TE insertions is the results of integration preferences and subsequent selective biases. For instance, it has been established that COPIA and MuDR transposons integrate preferentially within or upstream genes in multiple species. The effect of integration preference is particularly important in this article, as most TE insertion polymorphisms detected singletons that are expected to be the result of very recent transposition events unfiltered by natural selection.

Our sentences were poorly worded. We fully agree that selection is not the only factor here. We have now revised the wording and cite some relevant studies showing the breadth of scenarios.

Line 205-213 The increased frequency of the helitron insertion at MFS could be the consequence of either positive selection of the TE insertion, a linked polymorphism (genetic hitchhiking) or else background selection on the haplotype. Better evidence for positive selection on the TE insertion itself could be obtained by scanning the region for selective sweeps based on SNPs and polymorphic TEs and using site frequency spectrum-based (E.g Fst) and/or haplotype length-based (e.g. iHS). Selective sweeps signatures centered on the TE insertion would support selection on this variant.

This is an excellent suggestion. The different populations were in fact already scanned for iHS and XP-EHH based signatures. We now more explicitly state that this has been done and report the findings in for relevant TE insertion loci.

Line 225-227. The association between TE presence and azole resistance is tantalizing, but cannot imply causality. As pointed before, the TE insertions may be simply hitchhiking a causal mutation. Moreover, I cannot find in the article any result supporting that the TE insertions associated with azole resistance also show signatures of positive selection. Indeed, it seems that none of the insertion discussed at length in Lines 205-223 show associations in Suppl. Figure 12. This is a key point of the article because without this connection it is difficult to draw any conclusion about potential benefits associated with TE insertions in Zymoseptoria tritici.

Yes, we entirely agree that our manuscript does not provide conclusive evidence for adaptive functions of TE insertions. This was not our intention, and we regret the lack of clarity in our presentation. Evidence for adaptive TE insertions in the species was presented in different manuscripts but not here. We now more carefully discuss TE insertions which we consider candidates for having contributed adaptive variation and explicitly consider overlaps with selective sweeps.

Line 354-357. As I mentioned above, I don't see any correspondence between signatures of positive selection and azole resistance for the Gypsy (RLG_Sol) insertion near the MFS gene in Chr 12 (Figure S10). Indeed, this insertion is not marked as statistically significant in Figure S11.

We now more clearly mention this in the text.

Figure 6D. The reduction observed in the fraction of loci with low allele frequency (10%) in North American isolates is not accompanied by an increase in the proportion of loci at high allele frequency. In other words, it seems as if the bars do not sum up 1. Please check/clarify this figure.

We have re-checked the data presented and improved the visualization.

Work in Arabidopsis demonstrated that intraspecific genome size variation is largely explained by copy number variation in 45S rRNA (Long et al. 2013 Nat. Genet). This work should be discussed in relation to your observations, particularly given that the de novo assembly of genomes using short-reads is unable to estimate copy number variation of rRNA or centromeres.

Thank for this helpful context of genome size variation. We now mention this in the Discussion as requested.